# Manipulation of time-dependent multicolour evolution of X-ray excited afterglow in lanthanide-doped fluoride nanoparticles

Lei Lei [1], Yubin Wang[1], Weixin Xu[1], Renguang Ye[1], Youjie Hua[1], Degang Deng[1], Liang Chen[1], Paras N. Prasad [2] ✉ & Shiqing Xu [1] ✉

External manipulation of emission colour is of significance for scientific research and applications, however, the general stimulus-responsive colour modulation method requires both stringent control of microstructures and continously adjustment of particular stimuli conditions. Here, we introduce pathways to manipulate the kinetics of time evolution of both intensity and spectral characteristics of X-ray excited afterglow (XEA) by regioselective doping of lanthanide activators in core-shell nanostructures. Our work reported here reveals the following phenomena: 1. The XEA intensities of multiple lanthanide activators are significantly enhanced via incorporating interstitial Na⁺ ions inside the nanocrystal structure. 2. The XEA intensities of activators exhibit diverse decay rates in the core and the shell and can largely be tuned separately, which enables us to realize a series of core@shell NPs featuring distinct time-dependent afterglow colour evolution. 3. A core/multi-shell NP structure can be designed to simultaneously generate afterglow, upconversion and downshifting to realize multimode time-dependent multi-colour evolutions. These findings can promote the development of superior XEA and plentiful spectral manipulation, opening up a broad range of applications ranging from multiplexed biosensing, to high-capacity information encryption, to multidimensional displays and to multifunctional optoelectronic devices.

External manipulation of emission colour is of significance in the emerging fields of multiplexed biosensing[1,2], high-capacity information encryption[3,4], multidimensinoal displays[5,6], and multifunctional optoelectronic devices[7,8]. The general exploited strategy is the modulation of external stimuli conditions, such as excitation source, temperature, pressure, electric or magnetic field, for luminescent nanomaterials with purposefully designed structures[9–13]. However, in these cases, complex microstructures are stringently controlled, and special stimulus is required to be dynamically modulated. Manipulating time-dependent multicolour evolution without any external

stimulus is an ideal route to avoid these issues, but it is remaining an open challenge in inorganic systems.

Afterglow is an optical phenomenon that releases stored energy in traps as light after excitation ceases[14,15], whose radiative property changes gradually over time. Despite significant achievements made in tuning afterglow intensity, duration, and wavelengths mainly through modifying chemical compositions[16–18], it is hard to realize time-dependent afterglow colour evolution owing to the difficulty in blocking energy transfer between different activators[19–22]. For most previously reported long persistent phosphors, such as aluminates[23,24],

[1]Key Laboratory of Rare Earth Optoelectronic Materials and Devices of Zhejiang Province, Institute of Optoelectronic Materials and Devices, China Jiliang University, Hangzhou 310018, People's Republic of China. [2]Institute for Lasers, Photonics, and Biophotonics and Department of Chemistry, University at Buffalo, State University of New York, Buffalo, NY 14260, USA. ✉e-mail: pnprasad@buffalo.edu; shiqingxu@cjlu.edu.cn

silicates[25,26], sulfides[27,28], and carbon dots[29,30], the afterglows exhibit a time-dependent intensity decrease, while the spectral profiles remain unchanged. In these systems, although the afterglow intensity decreases over time in contrast to photoluminescence which is stable, its output colour remains unchanged. Fluoride nanoparticles (NPs) with the characteristics of low phonon energy and high photostablity are superior for the incorporation of lanthanide ions with abundant ladder-like energy levels, which are broadly employed to generate upconversion (UC), downshifting (DS), and X-ray excited afterglow (XEA) emissions[31-35]. Thus, through constructing and editing core@shell architecture, a time-dependent multicolour evolution might be realized on demand.

In this work, we introduce two kinds of universal platforms to manipulate the time-dependent multicolour evolution (Fig. 1): 1. Fluoride core@shell NPs (Type I) which exhibit afterglow colour evolution and 2. Fluoride core@shell@shell NPs (Type II) which exhibit multimode afterglow-based colour evolution, together with photon UC and DS. Incorporating interstitial $Na^+$ ions inside the nanocrystal structure was employed as an innovative route to amplify the XEA intensities of the $NaLuF_4$: Gd/(Pr, Tb, Dy, or Sm) NPs. The formed interstitial $Na^+$ ions promote the generation of anion Frenkel defects upon X-ray irradiation, benefiting the formation of traps followed by the improved XEA. Moreover, through constructing a core@shell structure, the processes in the activators doped in the core and those in the shell can be enhanced separately, which enables us to realize a series of core@shell NPs featuring distinct time-dependent afterglow colour evolution. We also demonstrate the simultaneous achievement of bright XEA, UC, and DS for multimode time-dependent colour evolutions by control of excitation dynamics. This is a first demonstration of regioselective activators resulting in distinct time-dependent multicolour spectral evolution in a core@shell nanostructure.

## Results

### Spectroscopic study of afterglow intensification

The precursors molar ratio of [Na]/[RE] for the preparation of hexagonal $NaREF_4$ via the co-precipitation method was generally fixed at 2.5: 1, which had been broadly employed in previous literatures[36-39]. Our results revealed that the pure hexagonal $NaLuF_4$: Gd/Tb NPs were remaining achieved, when tuning the [Na]/[RE] ratio from 2.5, 5, 7.5 to 10 (Fig. 2a). It should be mentioned that the [F]/[RE] ratio of 3.75 was employed to inhibit the formation of any NaF impurity phase. Transmission electron microscopy (TEM) images (Fig. 2d–g) revealed these hexagonal prism NPs were dispersed well in cyclohexane. Histograms of size distributions (Supplementary Fig. 1) suggested these NPs possessed similar mean particle sizes of ~45 nm. Energy dispersive X-ray spectroscopy (EDS, Supplementary Fig. 2) result demonstrated the existence of $Gd^{3+}$ and $Tb^{3+}$ dopants in the $NaLuF_4$ matrix and inductively coupled plasma-optical emission spectroscopy (ICP-OES, Table 1) analysis suggested the actual $Gd^{3+}$ and $Tb^{3+}$ doping concentrations were about 16 and 14 mol%, respectively, which were consistent with the designed nominal contents.

X-ray excited optical luminescence (XEOL) and XEA intensities of the $NaLuF_4$: Gd/Tb NPs were enhanced by ~5.5 and ~11 times with increasing the [Na]/[RE] ratio from 2.5 to 10, respectively (Fig. 2b, c and Supplementary Fig. 3). The XEA duration was also evidently prolonged with an increase of the [Na]/[RE] ratio (Supplementary Fig. 4a); especially, the green colour could be clearly observed even after 500 s when the [Na]/[RE] ratio was increased to 10 (Supplementary Fig. 4b). With further increasing the [Na]/[RE] ratio to 12.5, the NaF phase emerged and the XEOL as well as XEA intensities decreased (Supplementary Fig. 5). With increasing the [F]/[RE] ratio from 3.75 to 5, although an NaF impurity phase was formed at the [Na]/[RE] ratio above 5 (Supplementary Fig. 6), similar [Na]/[RE] ratio dependent XEOL and XEA intensities variation trends were observed as well

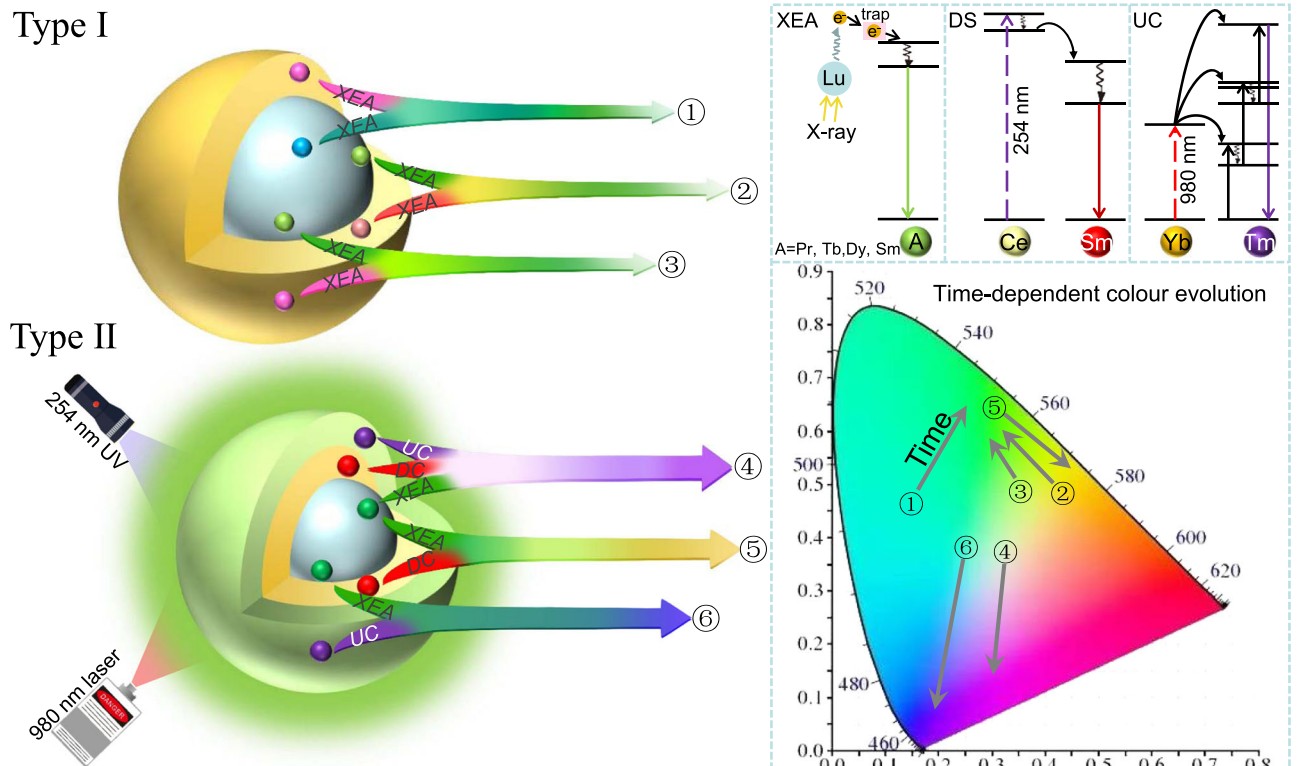

**Fig. 1 | Schematic illustration of the proposed two kinds of universal platforms for the realization of time-dependent multicolour evolutions.** In type I, the activators in the core and shell could be $Pr^{3+}$, $Tb^{3+}$, $Dy^{3+}$, or $Sm^{3+}$ ions. In type II, $Tb^{3+}$ ions doped in the core is used to generate green XEA and DS, Ce/Sm codopants in the first-shell layer are used to generate red DS emission, while Yb/Tm codopants in the second-shell layer are used to generate blue UC emission.

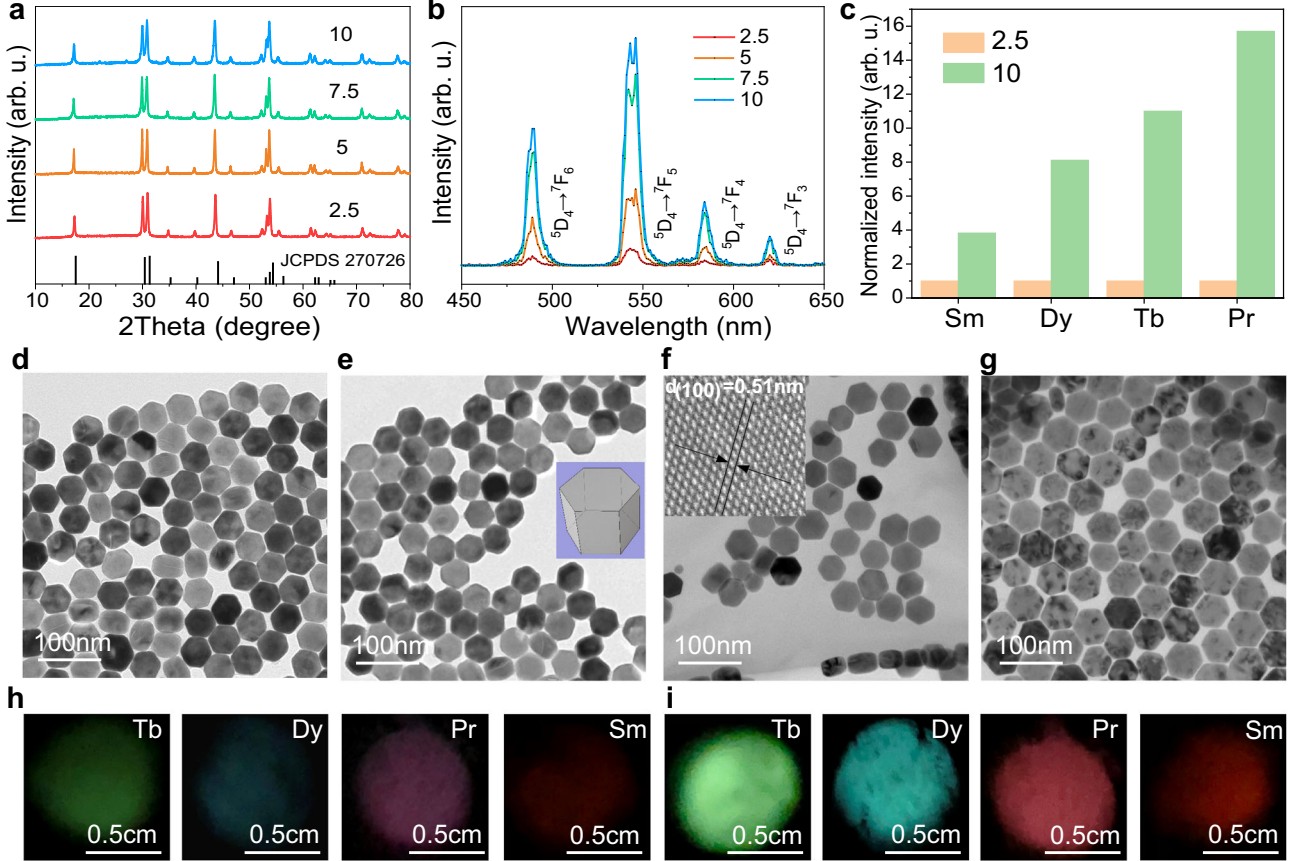

**Fig. 2 | Intensification of XEA via employing excessive Na⁺ precursors.** XRD patterns (**a**) and XEA spectra (**b**) of the NaLuF₄: Gd/Tb NPs prepared with different [Na]/[RE] ratios. **c** Integral XEA intensities of the NaLuF₄: 15Gd/(0.5Pr, 15Tb, 0.5Dy or 0.5Sm) NPs prepared with [Na]/[RE] of 2.5 and 10. The XEOL intensities for the [Na]/[RE] = 2.5 were normalized to 1. TEM images of the NaLuF₄: Gd/Tb NPs prepared with [Na]/[RE] of 2.5 (**d**), 5 (**e**), 7.5 (**f**), and 10 (**g**). Inset of **e** represents the three dimensional shape of a single NP. Inset of **f** shows the corresponding high-resolution TEM image. Photographs of those NPs doped with different activators at [Na]/[RE] of 2.5 (**h**) and 10 (**i**). X-ray operation was set at a voltage of 30 kV for 5 min. Source data are provided as a Source Data file.

(Supplementary Fig. 7). The removal of NaF does not change the relationship between the XEA intensity and the [Na]/[RE] ratio, i.e., the XEA intensity increased with the [Na]/[RE] ratio increasing from 2.5 to 10 and then decreased when the [Na]/[RE] goes up to 12.5 (Supplementary Fig. 8). It should be noted that the incorporation of $Gd^{3+}$ ions with an optimal concentration of 15 mol% was used to facilitate the population of $Tb^{3+}$ excited levels via energy transfer from $Gd^{3+}$ to $Tb^{3+}$ followed by the improved XEOL and XEA intensities (Supplementary Figs. 9, 10)[40,41]. With increasing the $Tb^{3+}$ concentration, the $Gd^{3+}$: $^6P_{7/2} \rightarrow {}^8S_{7/2}$ emission intensity decreases (Supplementary Fig. 11), revealing that $Gd^{3+}$ can promote the energy migration from its $^6P_{7/2}$ level to activators. Moreover, the relatively large energy gap (~$3.2*10^4$ cm⁻¹) between $^6P_{7/2}$ and the ground state of $Gd^{3+}$ in favor of minimizing energy loss caused by multiphonon assisted non-radiation relaxation and cross-relaxation[42].

The XEOL intensity of the NaLuF₄:Tb NPs ([Na]/[RE] = 10) is about 0.42 and 2.17 times the strength of the commercial CsI:Tl and BGO scintillators, respectively (Supplementary Fig. 12). The XEA intensity is stronger than that of previously reported $Sr_{13}Al_{22}Si_{10}O_{66}$:Eu[43] and commercial $SrAl_2O_4$:Eu/Dy, ZnS:Cu, ZnS:Cu/Co and CaS:Eu persistent phosphors (Supplementary Fig. 13), and the afterglow time can last up to more than 6 h (Supplementary Fig. 14). Moreover, the XEA intensity of $Tb^{3+}$ ions could be further intensified by ~2.2 times through coating of an inert NaYF₄ shell (Supplementary Fig. 15), which is attributed to the inhibition of energy migration from $Tb^{3+}$ to surface quenchers. Furthermore, the XEOL and XEA of the $Pr^{3+}$, $Sm^{3+}$, and $Dy^{3+}$ activators were also greatly improved with an increase of the [Na]/[RE] (Fig. 2c and Supplementary Figs. 16–18). The phenomenon of the intensified XEA were clearly observed from their corresponding photographs (Fig. 2h, i).

## Mechanistic investigation

ICP-OES results suggested that the actual [Na]/[RE] ratio in the NaLuF₄: Gd/Tb NPs increases from 1.096 to 1.312 with an increase of the nominal ratio from 2.5 to 10 (Table 1). The XRD peaks shifted toward lower angles (Fig. 3a) at higher [Na]/[RE] ratio, revealing the expansion of the crystal lattice. Rietveld XRD refinement results revealed that the crystal lattice increased from 5.9694 Å *5.9694 Å *3.5026 Å to 5.9762 Å *5.9762 Å *3.5154 Å (Supplementary Fig. 19). Thus, the interstitial Na⁺ ions probably are introduced in the crystal structure (Supplementary Fig. 20), and its content increases at higher [Na]/[RE] ratio. The charged oleate ligands coated on the NPs surface act as anionic species

**Table 1 | Nominal and ICP-OES results of the NaLuF₄: Gd/Tb NPs prepared with different [Na]/[RE]. [RE] = [Lu] + [Gd] + [Tb] = 100%**

| Nominal | ICP-OES results (mol %) | | | | Actual |
|---|---|---|---|---|---|
| [Na]/[RE] | [Lu] | [Gd] | [Tb] | [Na] | [Na]/[RE] |
| 2.5 | 69.9% | 16.4% | 13.7% | 109.6% | 1.096 |
| 5 | 68.8% | 17.6% | 13.6% | 119.1% | 1.191 |
| 7.5 | 70.5% | 15.1% | 14.4% | 125.6% | 1.256 |
| 10 | 68.0% | 17.4% | 14.6% | 131.2% | 1.312 |

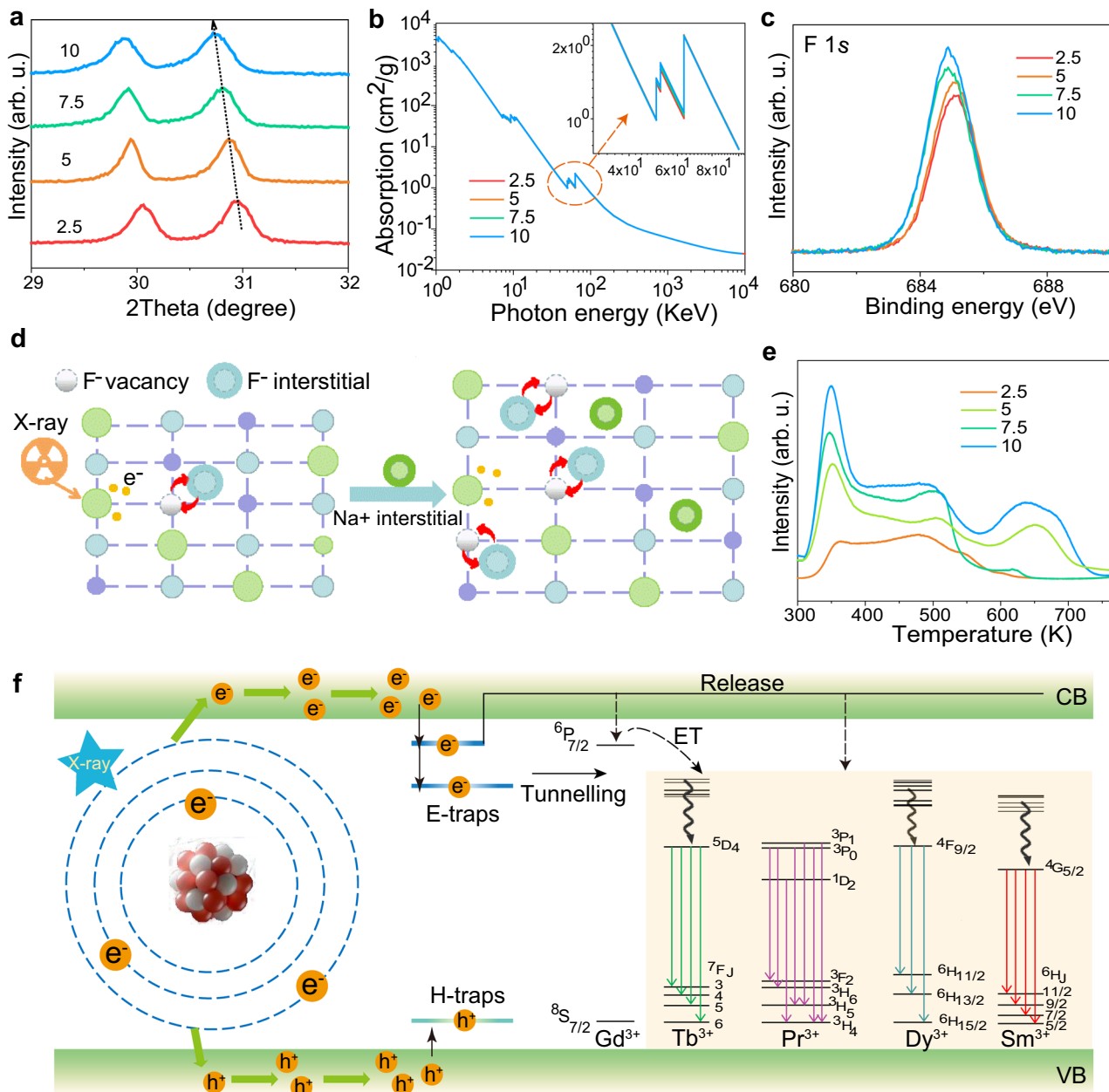

**Fig. 3 | Mechanistic investigations of the improved XEOL and XEA. a** Zoom-in XRD patterns of the NaLuF$_4$: Gd/Tb NPs prepared with different [Na]/[RE] ratio from 29 to 32°. **b** Absorption spectra of the NaLuF$_4$: Gd/Tb NPs prepared with different [Na]/[RE] ratio as a function of X-ray energy. XPS spectra of F 1$s$ (**c**). **d** Schematic illustration for increased Frenkel defects after the introduce of Na$^+$ interstitial upon X-ray irradiation. **e** TL spectra of the NaLuF$_4$: Gd/Tb NPs prepared with different [Na]/[RE]. **f** Proposed energy transfer mechanism for the high [Na]/[RE] ratio induced an increase of XEOL and XEA intensities. CB conduction band, VB valence band, ET energy transfer. Source data are provided as a Source Data file.

to maintain the charge balance[44,45]. X-ray energy dependent absorption spectra revealed the absorption coefficient of the NaLuF$_4$: Gd/Tb NPs prepared with different [Na]/[RE] ratio did not change (Fig. 3b). X-ray photoelectron spectroscopy (XPS) results indicated the binding energies of the F 1$s$ and Lu 4$d$ decreased at higher [Na]/[RE] ratio (Fig. 3c and Supplementary Fig. 21). Moreover, the formation of interstitial Na$^+$ ions will bring electrostatic interactions between the F$^-$ and interstitial Na$^+$ ions. Upon X-ray irradiation, both the reduced binding energy and introduced electrostatic interaction can promote the formation of anion Frenkel defects. Density functional theory (DFT) calculations revealed that the anion Frenkel defect formation energies ($E_f$) for the dislocation of F$^-$ ions into interstitial sites with different separation distances (0.5, 1.0, 1.5, 2.0, and 2.5 Å) were consistently reduced after the formation of interstitial Na$^+$ ions

(Supplementary Fig. 22a–c). Similar results were achieved upon changing the interstitial Na$^+$ sites (Supplementary Fig. 22d, e). As a result, in the case of high [Na]/[RE] ratio, anion Frenkel defects are easier to be formed via elastic collisions between large-momentum X-ray photons and small fluoride ions (Fig. 3d), benefiting the formation of traps[46]. X-ray activated thermoluminescence (TL) glow curves revealed that the shallowest trap depth was almost not changed with an increase of the [Na]/[RE] ratio (Fig. 3e), since the trap depth is generally proportional to the peak temperature. The TL intensity enhances at higher [Na]/[RE] ratio, suggesting the amount of shallowest traps is increased.

Based on the above analysis, a possible mechanism for the high [Na]/[RE] ratio induced increase of XEOL and XEA intensities is proposed (Fig. 3f). The input X-ray radiation energy interacts with heavy

lanthanide $Lu^{3+}$ ions to generate hot electrons and deep holes through the photoelectric effect and Compton scattering effect[3,47,48]. Then massive secondary electrons are generated via electron–electron scattering and Auger process, which leads to the production of charge carriers with low kinetic energy. These tremendously amounts of charge carriers are largely transported toward luminescence centers, followed by the XEOL emission, or captured by traps to produce subsequent afterglow emission. The spontaneous XEA studied in our case is generated via thermal disturbance at room temperature, which is mainly originated from the release of charge carriers deposited in the shallowest traps corresponding to the peak temperature at 350 K. With an increase of the [Na]/[RE] ratio, the concentration of shallowest traps increases, thus, the amounts of deposited charge carriers are significantly increased upon X-ray irradiation, which is subsequently followed by the evidently improved XEA intensity. Meanwhile, the traps located between the conduction band and the excited levels of the activators, can act as bridge centers to facilitate the electrons population in the excited levels. As a result, the XEOL intensity enhances with an increase of the [Na]/[RE] ratio as well. For the [Na]/[RE] ratio above 10, the non-active NaF reduces the total number of emitters and the excessive anion Frenkel defects might promote non-radiative relaxation. Both effects contribute to the decrease of the XEOL and XEA intensities.

To further verify that the high [Na]/[RE] ratio benefits the improvement of both XEOL and XEA of activators in the shell layer as well, a same $NaYF_4$ inert-core was employed to grow $NaLuF_4$:15Gd/15Tb active-shell NPs with similar size distributions at the [Na]/[RE] ratio of 2.5 and 10. As shown in Supplementary Figs. 23, 24, both of the achieved inert-core@active-shell NPs showed a pure hexagonal phase and similar mean particle size of ~62 ± 2 nm. As anticipated, the XEOL and XEA were evidently enhanced about 4 and 48 times, respectively (Supplementary Fig. 25), manifesting that the performances of XEOL and XEA can be significantly improved by increasing the [Na]/[RE] ratio in the shell precursors.

**Time-dependent afterglow colour evolution**

The above results reveal that both the XEA intensities of activators in the core and the shell can be modified by tuning the [Na]/[RE] ratio or the $Gd^{3+}$ doping content in the core or the shell layer. It is well known that the deleterious cross-relaxations between activators can be largely inhibited via spatially separating them in a core@shell nanostructure[49–51]. In this case, the relative intensities of activators in the core and shell could be tuned to satisfy a specific requirement. Moreover, the afterglow decay rate was independence of the [Na]/[RE] and $Gd^{3+}$ doping content (Fig. 4a, b), but highly related to the activators type (Fig. 4c). Evidently, the XEA intensity of the $Sm^{3+}$ ions decreased much faster than that of the $Tb^{3+}$ ions. As a result, by selectively incorporating $Tb^{3+}$ and $Sm^{3+}$ ions into the core and the shell separately, the output colour will be changed over time owing to the fast fading of the red component from $Sm^{3+}$ than the green one from the $Tb^{3+}$ ions (Supplementary Fig. 26). As shown in Supplementary Fig. 27, the XEA decay profile is independent of the Ln doping concentration for the Tb/Dy/Pr doped samples, indicating that the different afterglow decay rates of lanthanide activators are mainly ascribed to their different intrinsic arrangements of 4f electrons[52], instead of concentrations.

Theoretically, the output colour of XEA is facile to be modulated through adopting dual activators with tunable initial relative XEA intensities in a designed core@shell architecture. As a proof-of-concept, three kinds of core@shell NPs doped with different activators were studied. As shown in Fig. 4d, characteristic emissions of both $Tb^{3+}$ and $Sm^{3+}$ ions were observed in the XEOL spectra of the $NaLuF_4$:15Gd/15Tb@ $NaLuF_4$:10Gd/0.5Sm (named as Tb@Sm) NPs. The initial XEA colour was bright olivine, which then turned to green after 60 s (Fig. 4g and Supplementary Fig. 28). After substituting $Sm^{3+}$ with the $Pr^{3+}$ ions in the shell, the XEOL of both $Tb^{3+}$ and $Pr^{3+}$ ions were achieved

simultaneously as well (Fig. 4e). The XEOL intensity ratio of $Tb^{3+}$ to $Pr^{3+}$ was evidently increased by elevating the $Gd^{3+}$ ions doping concentrations in the core layer (Supplementary Fig. 29). The XEA colour of the $NaLuF_4$: 15Tb@ $NaLuF_4$:15Gd/0.5Pr (named as Tb@Pr) NPs was changed from yellow to green within an interval of 60 s (Fig. 4h and Supplementary Fig. 30). Similarly, the XEOL from both $Sm^{3+}$ and $Dy^{3+}$ ions were presented in the $NaLuF_4$:15Gd/0.5Sm@$NaLuF_4$:15Gd/0.5Dy (named as Sm@Dy) NPs (Fig. 4f). The corresponding XEA colour was changed from turquoise, to dark turquoise to green over time (Fig. 4i and Supplementary Fig. 31). Furthermore, a designed pattern was prepared by stamping the different core and core@shell NPs suspensions on an A4 paper (Fig. 4j). As shown in Fig. 4k, l, after the cessation of X-rays, the output colours remained the same over time when employing the core NPs as inks, while it exhibited evident time-dependent colour variations by employing the core@shell NPs. The XEA intensities were greatly reduced when codoping dual-activators (Tb/Sm, Tb/Pr, or Sm/Dy) in the core layer (Supplementary Fig. 32), indicating the spatial separation of activators in a core@shell structure is of importance for the realization of bright multicolour evolution. It should be noted that the physical mixture of different NPs will lead to the non-uniform colour distribution (Supplementary Fig. 33). These results suggest that the present developed core@shell NPs show promising applications in optical information storage and high-level anti-counterfeiting.

**Time-dependent colour evolution under different excitations**

Different from most reported long persistent phosphors, such as aluminates[23,24], silicates[25,26], sulfides[27,28] and carbon dots[29,30], the fluoride NPs are much more appropriate for the realization of simultaneously afterglow, DS and UC emissions via the construction of an appropriate core@hell nanoarchitecture. Through a judicious design of fluoride-based core/multi-shells NPs, time-dependent colour evolution can be manipulated by altering the pumping lights owing to the different electrons population pathways in the excited states of activators[3]. As a proof-of-concept, the $NaLuF_4$:15Gd/15Tb@Na$LuF_4$:15Gd/10Ce/0.5Sm@$NaGdF_4$:49Yb/1Tm (named as Tb@Ce/Sm@Yb/Tm) core@shell@shell NPs (Supplementary Fig. 34) were used to prepare a bowknot gel (Fig. 5a) to show the colour editing process (Supplementary Figs. 35, 36). Upon X-ray irradiation, the bowknot exhibited bright green afterglow corresponding to the $Tb^{3+}$: $^5D_4 \rightarrow {}^7F_j$ transitions (Fig. 5b). It should be noted that the $Ce^{3+}$ ions were employed to absorb the 254 nm UV photons and inhibit $Sm^{3+}$ XEA (Supplementary Fig. 37), and the $Yb^{3+}$ ions were used to absorb the 980 nm NIR photons and inhibit $Tm^{3+}$ XEA (Supplementary Fig. 38). After exchanging the $NaLuF_4$:15Gd/15Tb and $NaGdF_4$:49Yb/1Tm layer in the core@shell@shell structure, the UC intensity was enhanced while the XEA intensity was greatly reduced (Supplementary Fig. 39). This bowknot presented medium orchid UC corresponding to the $Tm^{3+}$ emission under 980 nm laser excitation, while it changed to sulfur yellow DS corresponding to the $Tb^{3+}$ and $Sm^{3+}$ emissions under UV illumination (Fig. 5b and Supplementary Fig. 40). When the 980 nm laser and UV lamp were used simultaneously, the colour changed to pink (Fig. 5b).

Benefiting from the above excitation-dependent multicolour emission characteristics, the time dependent colour evolution was successfully edited through integrating XEA whose intensity decreased gradually over time and stable UC and/or DS. As shown in Fig. 5c and Supplementary Fig. 41a, the pre-X-ray-irradiated bowknot gel exhibited bright green colour under 980 nm laser excitation, which then evidently changed to medium aquamarine, dark cyan, and slate blue gradually. Similarly, upon illuminating with a 254 nm UV lamp, the output colour evolved from green, to pale green, to khaki and then to light goldenrod yellow gradually (Fig. 5d and Supplementary Fig. 41b). Moreover, when the 980 nm laser with appropriate power was used along with the 254 nm UV lamp, the emission colour switched from

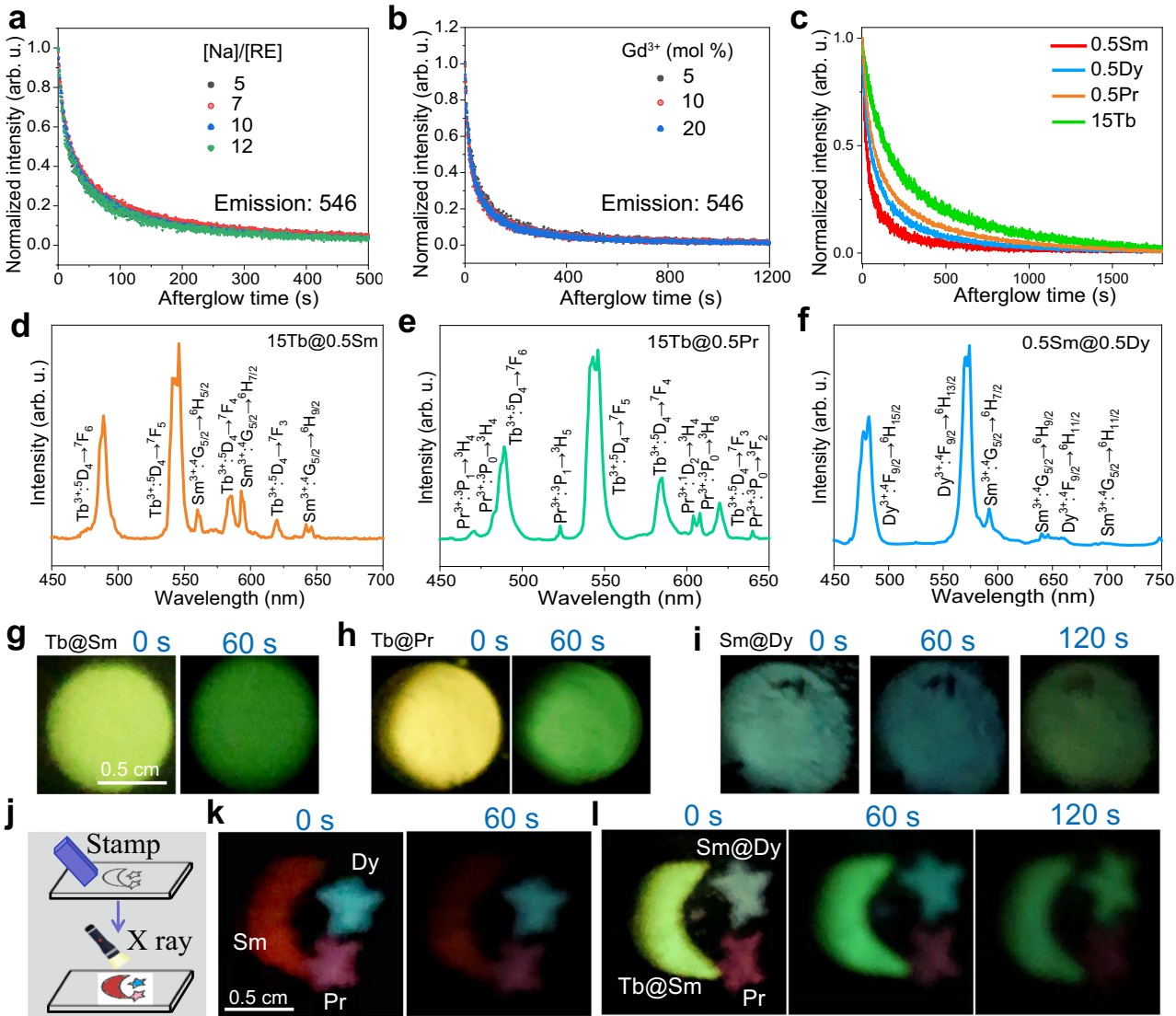

**Fig. 4 | Engineering time-dependent multicolour afterglow evolution via editing core@shell architecture. a** Normalized XEA decay curves of the NaLuF₄: 15Gd/15Tb core-only NPs prepared with different [Na]/[RE]. **b** Normalized XEA decay curves of the NaLuF₄: Gd/15Tb core-only NPs doped with different Gd³⁺ ions concentrations. **c** Normalized XEA decay curves of the NaLuF₄: 15Gd/(0.5Pr, 15Tb, 0.5Dy or 0.5Sm) core-only NPs. XEOL spectra and corresponding XEA photographs with different delay times of the Tb@Sm (**d, g**), Tb@Pr (**e, h**), and Sm@Dy (**f, i**) core@shell NPs. **j** Schematic showing the preparation of a pattern. An oak with a designed pattern was immersed into the NPs suspensions and then stamped on an A4 paper. **k** Afterglow photographs of the pattern by employing the NaLuF₄:15Gd/0.5Pr, NaLuF₄:15Gd/0.5Sm, and NaLuF₄:15Gd/0.5Dy core-only NPs suspensions. **l** Afterglow photographs of the pattern by employing the NaLuF₄:15Gd/0.5Pr core-only, Tb@Sm, and Sm@Dy core@shell NPs suspensions. The pattern was irradiated by an X-ray source at 30 KV for 5 min. Source data are provided as a Source Data file.

thistle, to plum, to light violet, and finally to medium violet over time (Fig. 5e). As a result, the XEA, DS, and UC emission colours are facile to be modulated by incorporating suitable activators into the fluoride-based core/multi-shells NPs, which might provide a universal strategy for the regulation of time-dependent colour evolution.

As an example of practical multidimensional display application, an ink painting was drawn by these core@shell NPs with time-dependent XEA colour evolution characteristic and other typical fluoride NPs. As shown in Fig. 6a, the Tb@Ce/Sm@Yb/Tm, Tb@Pr, NaLuF₄:Yb/Tm core-only NPs ink was used to draw the cloud, tree and boat, respectively. The NaLuF₄:Yb/Ho and NaLuF₄:Pr core-only NPs were used to plot the moon, while a mixture of Na₃HfF₇:Yb/Er and NaLuF₄:Yb/Ho core-only NPs was used to draw the lake (Supplementary Fig. 42). After the cessation of X-rays, the green cloud, red moon, yellow tree, and cyan lake were clearly observed, while their brightness became weaker and the tree turned to green after 30 s (Fig. 6b). When the 980 nm laser and 254 nm UV lamp were simultaneously used as

excitation sources, the cloud was changed from green to red, the moon was changed from red to yellow, the tree was changed from yellow to green, the lake was changed from cyan to red, and the blue boat emerged (Fig. 6b).

## Discussion

In conclusion, we have experimentally demonstrated amplification of XEA intensities of multiple lanthanide activators (Pr, Tb, Dy, or Sm) in fluoride NPs by incorporating interstitial Na⁺ ions inside the nanocrystal structure. The formed interstitial Na⁺ ions promote the generation of anion Frenkel defects when the NPs are exposure to X-rays, which promotes the formation of effective traps followed by improved XEA. Through employing a core@shell nanostructure, the XEA of activators doped in the core and the shell can be separately enhanced on-demand, enabling the editable time-dependent afterglow colour evolution. The intensified XEA is also employed for simultaneous realization of time-dependent multicolour UC and DS emissions. These

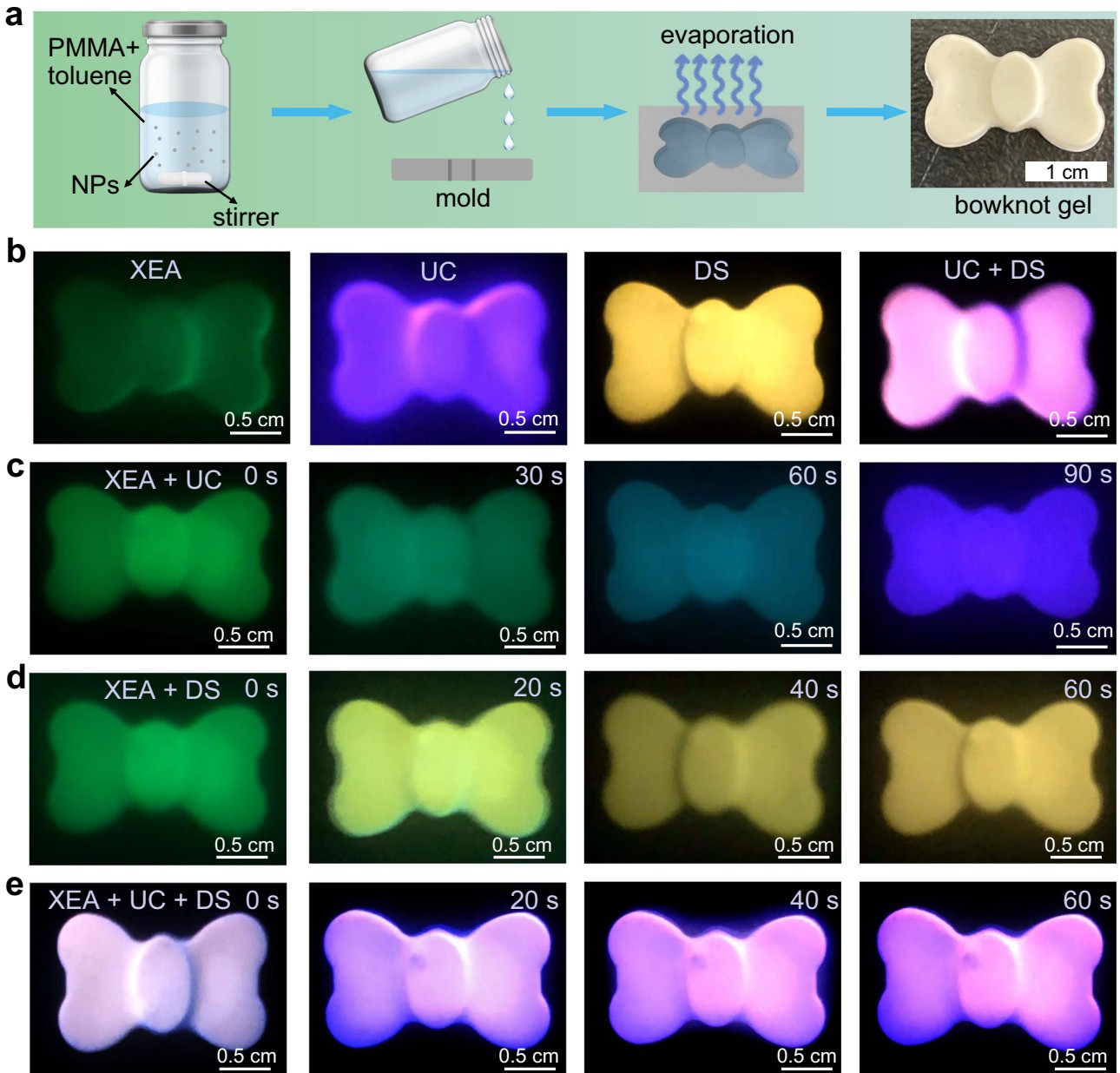

**Fig. 5 | Demonstration of time-dependent colour editing using fluoride-based core@shell@shell NPs. a** Schematic illustration for the synthetic procedures of bowknot gel. **b** Photographs of the bowknot under different excitation conditions. X-rays: 20 KV, 5 min. Time-dependent photographs of the pre-X-ray-irradiated bowknot gel, **c** under 980 nm laser excitation, X-rays: 30 KV, 5 min, **d** under 254 nm UV illumination, X-rays: 35 KV, 5 min, **e** under 254 nm UV lamp and 980 nm laser excitation.

results not only provide a facile avenue for developing X-ray activated afterglow nanomaterials with superior performances, but also to build a general platform to modulate the emission colour in fluoride core@shell NPs, which may find promising applications in biomedicine (i.e., photodynamic thereapy), advanced anti-counterfeiting, displays, optoelectronic devices (i.e., broadband photodectors) and potentially many others.

## Methods
### Chemicals and reagents
$Lu(Ac)_3·xH_2O$ (99.9%), $Gd(Ac)_3·xH_2O$ (99.9%), $Tb(Ac)_3·xH_2O$ (99.9%), $Dy(Ac)_3·xH_2O$ (99.9%), $Sm(Ac)_3·xH_2O$ (99.9%), $Pr(Ac)_3·xH_2O$ (99.9%), $Ce(Ac)_3·xH_2O$ (99.9%), $Yb(Ac)_3·xH_2O$ (99.9%), $Tm(Ac)_3·xH_2O$ (99.9%), $Ho(Ac)_3·xH_2O$ (99.9%), $Er(NO_3)_3·5H_2O$ (99.9%), $Yb(NO_3)_3·5H_2O$ (99.9%), $NH_4F$ (98%), $NaOH$ (≥96%), 1-Octadecene (ODE, 90%), Oleic acid (OA,

90%) and Hafnium (IV) acetylacetonate (97%) were supplied by Sigma Aldrich Company. Cyclohexane, methanol, and absolute ethanol were purchased from Sinopharm Chemical Reagent Company. Toluene was provided by the Hangzhou Gaojing Fine Chemical Industry Co., Ltd. Polymethyl methacrylate (PMMA) was purchased from Cool Chemical Technology (Beijing) Co., Ltd. All chemicals were of analytical grade and used as received without further purification.

### Synthesis of NaLuF$_4$:15Gd/15Tb NPs
$Lu(Ac)_3$ (0.56 mmol), $Gd(Ac)_3$ (0.12 mmol) and $Tb(Ac)_3$ (0.12 mmol) were added into a 50 mL three-necked bottle containing 8 mL OA. The mixture was heated at 150 °C for 30 min to remove water from the solution under $N_2$ atmosphere. Then 12 mL ODE was quickly added into the above solution and the resulted mixture was heated at 150 °C for another 30 min to form a transparent solution, and then cooled down

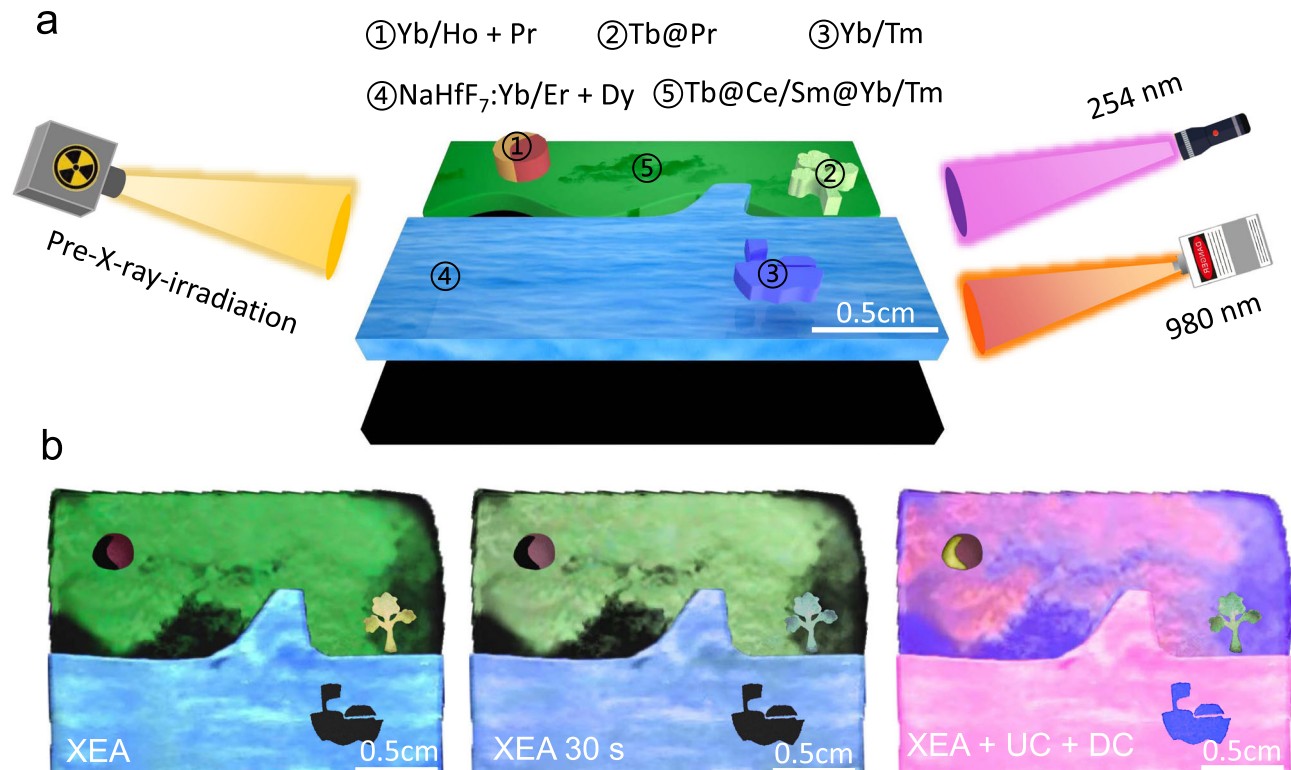

**Fig. 6 | Demonstration of XEA-based multidimensional display application.**
**a** Schematic illustration for the preparation of ink painting containing 5 layers.
**b** Photographs of the pre-X-ray-irradiated ink painting at 0, 30, and 980 nm/254 nm illumination. From left to right, both intensity and colour of the cloud, tree, boat, moon, and lake are changed.

to room temperature. Afterwards, 8 mL methanol solution containing $NH_4F$ (3, 4, 5 mmol) and NaOH (2, 4, 6, 8, and 10 mmol) was added into the above solution and stirred at 60 °C for 40 min. After the methanol was fully evaporated, the solution was heated to 300 °C under $N_2$ atmosphere and kept for 90 min, and then cooled down to room temperature. The products were precipitated by the addition of ethanol, collected by centrifugation, washed with ethanol, and finally dried or re-dispersed in 4 mL cyclohexane for further use. The other core-only NPs were prepared via similar experimental procedures, except with different doping activators.

**Synthesis of $NaLuF_4$:15Gd/15Tb@$NaLuF_4$:15Gd/10Ce/0.5Sm core@shell NPs**

$Lu(Ac)_3$ (0.596 mmol), $Gd(Ac)_3$ (0.12 mmol), $Ce(Ac)_3$ (0.08 mmol) and $Sm(Ac)_3$ (0.004 mmol) were added into a 50 mL three-necked bottle containing OA (8 mL). The mixture was heated at 150 °C for 30 min to remove water from the solution under $N_2$ atmosphere. A solution of ODE (12 mL) was then quickly added and the resulted mixture was heated at 150 °C for another 30 min to form a clear solution, and then cooled down to 80 °C. Thereafter, the pre-prepared $NaLuF_4$: 15Gd/15Tb core NPs in 4 mL cyclohexane was added to the above solution and kept at 110 °C for 40 min. After the removal of cyclohexane, 8 mL methanol solution containing $NH_4F$ (3 mmol) and NaOH (2 mmol) were added and the solution was stirred at 60 °C for 30 min. After the methanol was evaporated, the solution was further heated at 300 °C under $N_2$ for 120 min, and finally cooled down to room temperature. The other core@shell NPs were prepared via similar experimental procedures except using different doping activators and [Na]/[RE] ratio.

**Synthesis of $NaLuF_4$:15Gd/15Tb@$NaLuF_4$:15Gd/10Ce/ 0.5Sm@$NaGdF_4$:49Yb/1Tm core@shell@shell NPs**

$Gd(Ac)_3$ (0.4 mmol), $Yb(Ac)_3$ (0.392 mmol) and $Tm(Ac)_3$ (0.008 mmol) were added into a 50 mL three-necked bottle containing OA (8 mL).

The mixture was heated at 150 °C for 30 min to remove water from the solution under $N_2$ atmosphere. A solution of ODE (12 mL) was then quickly added and the resulted mixture was heated at 150 °C for another 30 min to form a clear solution, and then cooled down to 80 °C. Thereafter, the pre-prepared $NaLuF_4$:15Gd/15Tb@$NaLuF_4$:15Gd/ 0.5Sm core-shell NPs in 4 mL cyclohexane was added to the above solution and kept at 110 °C for 40 min. After the removal of cyclohexane, 8 mL methanol solution containing $NH_4F$ (3 mmol) and NaOH (2 mmol) were added and the solution was stirred at 60 °C for 30 min. After the methanol was evaporated, the solution was further heated at 300 °C under $N_2$ for 120 min, and finally cooled down to room temperature.

**Synthesis of lanthanide-doped $Na_3HfF_7$: 8Yb/6Er NPs**

Hafnium (IV) acetylacetonate (0.84 mmol), $Yb(NO_3)_3 \cdot 5H_2O$ (0.08 mmol), $Er(NO_3)_3 \cdot 5H_2O$ (0.06 mmol) and 20 mL ethanol were added into a 50 mL beaker and stirred for 10 min. Then 12 mL OA, 2.5 mL OM and NaOA (8.21 mmol) were added into the above mixture with continuous stirring for another 20 min followed by the addition of 5 mL deionized water containing 10 mmol $NH_4F$. After stirring at room temperature for about 30 min, the above solution was transferred into a 40 mL Teflon-lined autoclave, sealed and heated at 130 °C for 12 h.

**Synthesis of $NaLuF_4$:10Yb/10Ho NPs**

$Lu(Ac)_3$ (0.64 mmol), $Yb(Ac)_3$ (0.08 mmol) and $Ho(Ac)_3$ (0.08 mmol) were added into a 50 mL three-necked bottle containing OA (8 mL). The mixture was heated at 150 °C for 30 min to remove water from the solution under $N_2$ atmosphere. Then 12 mL ODE was quickly added into the above solution and the resulted mixture was heated at 150 °C for another 30 min to form a transparent solution, and then cooled down to room temperature. Afterwards, 8 mL of methanol solution containing $NH_4F$ (3 mmol) and NaOH (2 mmol) was added into the above solution and stirred at

**Table 2 | Comparison of different samples studied in this work**

| Sample | [Na]/[RE] | Size (nm) | Architecture | Emission mode | Emission color |
|---|---|---|---|---|---|
| $NaLuF_4$:15Gd/0.5Pr | 10 | / | Core-only | XEA | Pink |
| $NaLuF_4$:15Gd/15Tb | 10 | 47 | Core-only | XEA | Green |
| $NaLuF_4$:15Gd/0.5Dy | 10 | / | Core-only | XEA | Cyan |
| $NaLuF_4$:15Gd/0.5Sm | 10 | / | Core-only | XEA | Red |
| $NaLuF_4$:15Gd/15Tb@ $NaYF_4$ | Core 10 Shell 2.5 | / | Core@shell | XEA | Green |
| $NaYF_4$ | 2.5 | 49.5 | Core-only | / | / |
| $NaYF_4$ @$NaLuF_4$:15Gd/15Tb | Core 2.5 Shell 2.5 | 61.5 | Core@shell | XEA | Green |
| $NaYF_4$ @$NaLuF_4$:15Gd/15Tb | Core 2.5 Shell 10 | 64.5 | Core@shell | XEA | Green |
| $NaLuF_4$:15Gd/15Tb@ $NaLuF_4$:15Gd/0.5Sm | Core 10 Shell 10 | / | Core@shell | XEA | Pale yellow → Green |
| $NaLuF_4$:15Gd/15Tb@ $NaLuF_4$:15Gd/0.5Pr | Core 10 Shell 10 | / | Core@shell | XEA | Yellow → Green |
| $NaLuF_4$:15Gd/0.5Sm@ $NaLuF_4$:15Gd/0.5Dy | Core 10 Shell 10 | 60 | Core@shell | XEA | Turquoise → Dark turquoise → Green |
| $NaLuF_4$:15Gd/15Tb@$NaLuF_4$:15Gd/10Ce/0.5Sm@$NaGdF_4$:49Yb/1Tm | Core 10 Shell1 2.5 Shell2 2.5 | 90 | Core@shell@shell | XEA | Green |
|  |  |  |  | DC | Orange |
|  |  |  |  | UC | Purple |
| $NaLuF_4$:10Yb/10Ho | 2.5 | 31 | Core-only | UC | Yellow |
| $NaLuF_4$:49Yb/1Tm | 2.5 | / | Core-only | UC | Purple |

60 °C for 40 min. After the methanol was fully evaporated, the solution was heated to 290 °C under $N_2$ atmosphere and kept for 90 min, and then cooled down to room temperature.

The characteristics of the studied NPs were summarized in Table 2.

### Preparation of bowknot gel

Polymethyl methacrylate (PMMA, 100 mg) and 1 mL toluene were mixed, which then were stirred to form a transparent solution. 50 mg core@shell@shell NPs was added into the above PMMA toluene solution, which then was stirred for more than 3 h to generate uniform paste product without agglomerate. The resultant product was transferred into a designed mold drop by drop, which then volatilized at room temperature for 12 h until the bowknot was formed.

### Characterizations

X-ray diffraction (XRD) analysis was carried out by a powder diffractometer (Bruker D8 Advance) with a Cu-Kα (λ = 1.5405 Å) radiation. The morphology and size of the products were characterized by a field emission transmission electron microscopy (TEM, FEI Tecnai $G^2$ F20) equipped with an energy dispersive X-ray spectroscopy (EDS, Aztec X-Max 80T). XPS analyses were performed using Thermo Fisher Scientific K-Alpha with tube voltage 15 kV and tube current 10 mA. Inductively coupled plasma (ICP) analyses were performed using Agilent Varian 720. The DS and UC emission spectra were recorded with a spectrometer (Edinburgh FLS980) equipped with an adjustable laser diode (980 nm, 0–2 W) and Xenon lamp (450 W). Scintillation spectra were recorded on an OmniFluo-Xray-JL system (PMT-CR131-TE detector, 185–900 nm) with a mini MAGPRO X-ray excitation source. X-ray absorption spectra were obtained with a BL14W beamline at the Shanghai Synchrotron Radiation Facility, where the storage rings were operated at 3.5 GeV with a stable current of 200 mA. The data was recorded by a Lytle detector coupled with Si (111) double-crystal monochromator. To measure the thermoluminescence (TL) glow curves, the NPs were first exposed to the X-ray source for 15 min at room temperature. After the cessation of X-rays, the TL signals were recorded with a photomultiplier tube detector (Hamamatsu, R928P) and a spectrometer (Ocean Optics, QE Pro). The temperature was controlled by a heating stage, from RT to 500 °C with a rate of 1 °C/s.

### Computation method

The 2 × 2 × 1 supercell of $NaLuF_4$ was employed for the density functional theory (DFT) calculation. The structural optimizations and electronic structure calculations were carried out in the framework of the DFT with the projector augmented plane-wave method, as implemented in the Vienna ab initio simulation package (VASP)[53]. The generalized gradient approximation proposed by Perdew–Burke–Ernzerhof (PBE) was chosen for the exchange-correlation potential[54]. The cut-off energy for a plane wave was set to 500 eV. The energy criterion was set to $10^{-5}$ eV in the iterative solution of the Kohn–Sham equation. The Brillouin zone integration was performed using a 3 × 3 × 9 k-mesh. All the structures were relaxed until the residual forces on the atoms are less than 0.01 eV/Å. The anion Frenkel defect formation energies $E_f$ for the dislocation of $F^-$ ions into interstitial sites with different distances (0.5, 1.0, 1.5, 2.0, and 2.5 Å) were calculated. The $E_f$ values for the conditions of different interstitial $Na^+$ sites were calculated with considering the relaxation of all atoms owing to the introducing of interstitial $Na^+$ ions and formation of Frenkel defects. The separation distances in the four cases are no more than 0.25 Å.

### Reporting summary

Further information on research design is available in the Nature Research Reporting Summary linked to this article.

## Data availability

The authors declare that the data that support the findings of this study are available within the article and its Supplementary Information files. The raw data generated in this study are provided in the Source Data file. Source data are provided with this paper.

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

## Acknowledgements

This work is supported by Zhejiang Provincial Natural Science Foundation of China (LZ21A040002), National Natural Science Foundation of China (No. 52172164, 51872270), National Natural Science Foundation of China Joint Fund Project (U190920054). The Institute for Lasers, Photonics and Biophotonics acknowledges support from the office of Vice President for Research and Economic Development at the University at Buffalo.

## Author contributions

L.L. and Y.W. initiated and designed the project. Y.W. and W.X. performed nanoparticle synthesis. Y.W. and W.X. conducted optical measurements and taken photographs. L.L., Y.W., P.P., and S.X. wrote and revised the manuscript. L.L., Y.W., R.Y., Y.H., D.D., L.C., P.P., and S.X. contributed to the data analyses and discussion. These authors contributed to this work equally: L.L., Y.W., and W.X.

## Competing interests

The authors declare no competing interests.
