## [Peer Review File · Nature Communications]

Manipulation of time-dependent multicolor evolution of X-ray excited afterglow in lanthanide-doped fluoride nanoparticlesREVIEWER COMMENTS

Reviewer #1 (Remarks to the Author):

The paper describes a detailed investigation on X-Ray excited afterglow in lanthanide doped nanoparticles. The experimental work is very well conducted and the results are reliable and interesting for the scientific community. Nonetheless, in my opinion the topic and the proofs of concept highlighted in the paper are very similar to those reported by Zhuang et al. *Light: Science & Applications* (2021)10:132, which is also focused on X-ray-charged bright persistent luminescence in NaYF₄:Ln³⁺@NaYF₄ nanoparticles for multidimensional optical information storage. Although in the submitted paper the kind of fluoride materials are slightly different (NaLuF₄ instead of NaYF₄), a core@shell architecture and several combinations of luminescent lanthanide ions are similarly considered. Moreover, in the abovementioned paper (by Zhuang et al.), optical information-storage application is also demonstrated by inkjet based technology of the investigated nanoparticles. Therefore, I think that the paper does not contain a very high novelty to be considered for publication in *Nature Communications*. I suggest to submit to *Light: Science & Applications*.

Reviewer #2 (Remarks to the Author):

The manuscript "Manipulation and amplification of time dependent multicolour evolution of X-ray excited afterglow in lanthanide doped fluoride nanoparticles" by S. Xu and co-workers reports the synthesis and characterization of persistent nanophosphors based on lanthanide doped NaLuF₄ nanoparticles (a structure that is very similar to that reported by F. Zhang, *Nature Nanotechnology* 16 (9), 1011-1018). Adjustment of the [Na]/[RE] ratio as well as the dopant's concentration and positioning within core-(multi)shell structures allow to tune the X-ray induced afterglow color of the nanoparticles and adds additional downshifting and upconversion features. Combination of Ln dopants with different decay times and additional UV or NIR excitation results in time-dependent color modulation. These nanophosphors may be interesting for application in advanced anti-counterfeiting, as proposed by the authors. Additionally proposed applications include biomedicine, optoelectronic devices and displays – which is not fully clear to me and which should be elaborated in more detail (e.g. how would the time-dependent change of emission colour benefit biomedical application? how would a display the colour of which is determined by the decay time of the persistent phosphor be suitable for rapid color changes of pixels?). As such, the photophysics are very interesting, yet practical application potential, as highlighted by the authors, is vague.

Additional Comments and Questions:

- Incorporation of Na ions: the nominal ratio of [Na]/[RE] is significantly increased whereas the actual increase is very small (Table S1). How was the [Na] determined? Can such small increase in [Na] within the NPs indeed induce such strong XEA enhancement; any theoretical approach that would support this?
- What is the reason for the XEA loss at a ratio of 12.5?
- NaF formation: NaF can easily be washed away with a water-ethanol mixture. XEOL and XEA data were obtained on dry powders? If parts of these are indeed NaF, that may simply have an effect of how many X-ray-activated NPs there are and how many emitters compared to non-active NaF? How do the samples perform after washing to remove NaF?
- Can values for the afterglow (decay times) be given for comparison with phosphors reported in the literature (under comparable X-ray activation)? Are different afterglow time scales for various Ln due to the intrinsic Ln properties (references?) or due to the chosen mol%?
- Combination with various Ln (Figure 3): what is the role of Gd in the mechanism?
- Core-shell architectures: It is highlighted that XEA can be observed selectively from core or shell. Are core-shell structures truly needed? Would the same effect be observed when doping Ln ions for XEA together in the core? I am further wondering why the author chose to dope the UC emitters into the outer shell: this is the most sensitive location for upconverters given surface quenching; the core might have been the more intuitive region for UC dopants.
- Size and size distributions: standard deviations should be provided. Shell thicknesses should be

included.

- Figure panels: the figure panels are of high quality and attractive. I think they could be further improved if adding labels for used Ln ions or concentrations to allow the reader to identify these critical parameters at a glance rather than looking for them in captions or text.
- What are the dimensions of the phosphor spots photographed (a scale bar would be helpful)? Are those photographs of dry powders or dispersions? At which time point were the photos taken?
- Figure 2c: The linear lines may be misleading towards a linear correlation between ratio and intensity (which seems not to be the case based on intensities seen in Fig. 2b).
- Multi-color modulation: How fast do these color changes happen? Providing color points at shorter time intervals may be helpful. Figure 4: Spectra as a function of time to show the contributions of the different Ln may be insightful.
- XEA / DS / UC combination: What was the rationale to suppress XEA from Sm? Was there any EXA seen from Tm (as in F. Zhang, Nature Nanotech 2021)?
- DS and UC spectra should be given to show which Ln ions are active as a function of time as well as E-level diagram showing the excitation and emission pathways.
- Comparison of the performance of the nanophosphors with those reported in the literature is missing (while literature is discussed in the introduction, there is lack of discussion of the results in light of existing literature). I am further wondering how balanced the reference list is in terms of diversity of research groups and their geographical origin.
- There are a few typos and sentences that may need clarification:
 - o line 24 (the a)
 - o line 185 (produce)
 - o what is meant by “the afterglows exhibit same excitation and emission wavelengths”? Work by F. Zhang or B. Viana demonstrates examples of X-ray excited vis or NIR emission, clearly different excitation and emission wavelengths.
 - o “time-dependent color modulation on demand” – the color change seems to be dependent on the decay of the X-ray induced afterglow. How can this be controlled on demand?
 - o Figure S3: the caption should probably also include 12.5.
 - o Na₃HfF₇:Yb/Er and NaLuF₄:Yb/Ho are mentioned – how were these obtained and what are their sizes / size distributions?
 - o I could not find information about instrumentation used for DS and UC.

Reviewer #3 (Remarks to the Author):

This manuscript is concerned with the time dependent multicolour evolution of x-ray afterglow from lanthanide doped fluoride nanoparticles. This is mainly achieved by using different lanthanide ions as activators as they control the rate of afterglow decay. Then by having different compartments (core/shell) one has one part of the NP glowing in one color, the other in another color. By having the color intensity decay at different time scales leads to a changing color observed. This color effects of the NPs can be further modified by illuminating them with NIR or UV light, in order to trigger upconversion (UC) or downshifting (DS), respectively.

In general, this is interesting work and the authors also performed a comprehensive and careful characterization work on their nanoparticles and their photophysical properties. This was done for a systematic variation of the composition of the NPs, which is another strength of this article. Interesting here is certainly the observation that they can increase substantially the emission intensity by incorporating Na⁺ ions. In summary, a number of interesting optical properties are demonstrated in this work.

However, one crucial point regarding the chosen architecture of these core-shell or core-shell-shell nanoparticles still must be explained. This concerns the explanation, why the construction of such rather complex NPs is necessary at all to arrive at the reported phenomena. In principle, many of them should also be achievable by simply mixing the individual NP with the materials contained in the core and the shell (or shells), isn't that the case. Therefore, the central point to be made is to explain what is the gain from having these different materials contained within one more complex NP that requires correspondingly more synthetic effort.

In Figs. 4 and 5 the change of the colors is shown visually. Interesting here could also be to see the

full emission spectra as a function of time.

Interesting would also to know how high is the percentage of the energy of the light emitted by the NPs compared to the energy put in via x-ray radiation.

As a minor point on page the units of the crystal lattice should be given.

Finally, the use of the English language should be improved at various places, best by having this done by a native speaker.

In summary, I think that this manuscript has the potential for becoming published but the crucial point for that would be a convincing explanation, why the structure of the NPs employed here is superior to simply a mixture of individual NPs.

Response to reviewer's comments

We greatly appreciate the reviewers' insightful comments which are very helpful for improvement of our manuscript. In response to the valuable comments raised by the referees, we provide point-by-point responses along with the modifications (marked in blue) made in the revised manuscript.

Reviewer 1:

Comment: The paper describes a detailed investigation on X-Ray excited afterglow in lanthanide doped nanoparticles. The experimental work is very well conducted and the results are reliable and interesting for the scientific community. Nonetheless, in my opinion the topic and the proofs of concept highlighted in the paper are very similar to those reported by Zhuang et al. *Light: Science & Applications* (2021)10:132, which is also focused on X-ray-charged bright persistent luminescence in NaYF₄:Ln³⁺@NaYF₄ nanoparticles for multidimensional optical information storage. Although in the submitted paper the kind of fluoride materials are slightly different (NaLuF₄ instead of NaYF₄), a core@shell architecture and several combinations of luminescent lanthanide ions are similarly considered. Moreover, in the abovementioned paper (by Zhuang et al.), optical information-storage application is also demonstrated by inkjet based technology of the investigated nanoparticles. Therefore, I think that the paper does not contain a very high novelty to be considered for publication in *Nature Communications*. I suggest to submit to *Light: Science & Applications*.

Response: We appreciate the referee for this comment and we have read the suggested reference (*Light: Science & Applications* (2021)10:132) carefully. Actually, the reported literature is focused on XEA of single-type lanthanide ions doped NaYF₄:Ln³⁺@NaYF₄ NPs. In contrast, our work is devoted to developing a brand-new mechanism for the enhancement of XEA as well as novel spectral manipulation processes via multiple lanthanide ions doping in core@multishells NPs. Accordingly, a very creative time-dependent multicolour evolution has been successfully realized, and a mechanism for optical information-storage has been

proposed as well, which is distinct from that described in the suggested reference,. Detailed comparisons between the present work and the one in the suggested reference are given below.

Firstly, the suggested literature reported the XEA phenomenon in $\text{NaYF}_4:\text{Ln}^{3+}@\text{NaYF}_4$. In the present study, however, we developed a novel feasible method to greatly enhance the XEA intensities of multiply doped (Pr^{3+} , Tb^{3+} , Dy^{3+} and Sm^{3+}) nanophosphor by means of introducing interstitial Na^+ ions in both the NaLuF_4 core and the shell layer, considering that the enhancement in XEA intensity is of great importance for potential application of XEA in many areas such as bio-imaging and optical information storage. In addition, the time-dependent XEA colour evolution via judicious core/shell doping design enables new applications in anti-counterfeiting and bio-sensing. Therefore, the concept proposed in the present study should not be taken as that in the suggested literature. Instead, the strategy regarding core/shell doping design is actually a brand-new development towards manipulation of the XEA performance.

Secondly, the suggested literature mainly studied the XEA of NPs with a single Ln^{3+} activator. In contrast, systematical study on the incorporation of dual-activators into core-shell nanostructure has been performed in the present work in order to control time-dependent XEA spectral variations.

Thirdly, different strategies are adopted to demonstrate the application of XEA NPs in optical information storage. In the suggested literature, three different kinds of luminescent inks containing different $\text{NaYF}_4:\text{Ln}^{3+}@\text{NaYF}_4$ NPs ($\text{Ln} = \text{Tb}, \text{Dy}, \text{and Ho}$) were applied to print overlapping patterns for 3D optical information storage. In such case, optical filters have to be used for the colour variation. The present work, utilizes the various XEA decay times for different Ln activators located in both the core and the shell layers to realize time-dependent colour evolution in XEA, which can further be manipulated on demand

Finally, owing to the improved XEA performances of different lanthanide activators in the present work, the specially-designed core/multi-shells fluoride NPs can be employed to simultaneously realize XEA, UC and DS emissions. Therefore,

upon various excitation, one single nanostructure can exhibit several different time-dependent colour evolution modes, which may enable creative applications for optical information storage; this has not been reported in previous literatures.

Reviewer 2:

Comment: The manuscript “Manipulation and amplification of time dependent multicolour evolution of X-ray excited afterglow in lanthanide doped fluoride nanoparticles” by S. Xu and co-workers reports the synthesis and characterization of persistent nanophosphors based on lanthanide doped NaLuF₄ nanoparticles (a structure that is very similar to that reported by F. Zhang, Nature Nanotechnology 16 (9), 1011-1018). Adjustment of the [Na]/[RE] ratio as well as the dopant’s concentration and positioning within core-(multi)shell structures allow to tune the X-ray induced afterglow colour of the nanoparticles and adds additional downshifting and upconversion features. Combination of Ln dopants with different decay times and additional UV or NIR excitation results in time-dependent colour modulation. These nanophosphors may be interesting for application in advanced anti-counterfeiting, as proposed by the authors. Additionally proposed applications include biomedicine, optoelectronic devices and displays – which is not fully clear to me and which should be elaborated in more detail (e.g. how would the time-dependent change of emission colour benefit biomedical application? how would a display the colour of which is determined by the decay time of the persistent phosphor be suitable for rapid colour changes of pixels?). As such, the photophysics are very interesting, yet practical application potential, as highlighted by the authors, is vague.

Response: We thank the reviewer’s valuable comments and insightful suggestions. Taking into account this comment, the promising applications described in the last sentence of “Discussion” section are slightly modified to “biomedicine, advanced anti-counterfeiting, displays, optoelectronic devices and potentially many others”. To further elaborate these potential applications, more descriptions are added in the revised supporting information. **Biomedicine:** XEA can be used to monitor the drug release by coating drugs with absorption at specific wavelengths on the surface of NPs (XEA colour variation can vary before and after drug release). For example, after coating Doxorubicin or Daunorubicin (with absorption wavelength at a green region, Chem. Mater., 2017, 29, 7615-7628; Chem. Sci., 2018, 9, 3233) on the Tb@Sm or Tb@Pr NPs, XEA of the composites consists of mainly red emissions. Upon release

of the coated drugs, the XEA colour will change from red to green (Figure R1). Furthermore, since the XEA can be used to judge if a drug is released and the UC process can be used for bio-detection (ACS Nano, 2021, 15, 3709-3735; Biomaterials Advances, 2022, 136, 212763), a combination of XEA and UC might be used to guide the bio-detection. In addition, persistent phosphors show promising applications in bio-imaging and photodynamic therapy (Adv. Funct. Mater. 2018, 1707496; Nature Nanotechnology, 2021, 16, 1011–1018; ACS Nano, 2019, 13, 10419–10433; Nano Lett. 2019, 19, 8234–8244; Nano Energy, 2021, 79, 105437). In our work, the improved XEA performance of the lanthanide doped fluoride NPs is believed to facilitate their application in bio-imaging. **Anti-counterfeiting:** Multicolour NPs for advanced anti-counterfeiting has been verified in our work as well as in many other reports (Nature Communications, 2017, 8, 899; Adv. Mater. 2019, 1901430; Small, 2020, 2000708; Adv. Funct. Mater. 2021, 31, 2009920; Adv. Optical Mater. 2019, 1900519). **Display:** By increasing the X-ray irradiation power or time, the afterglow time can be prolonged to hours in our case (Fig. S14). Moreover, in the previous reported literature (Nature, 2021, 590, 410), the XEA of lanthanide doped NPs can last more than 30 days. Thus, through integrating XEA with UC and DS, the emission color switch can be feasibly achieved by tuning the NIR or UV excitation power with controlled time intervals. The output colour can be kept changed in hours, which could find applications for display. **Optoelectronic devices:** the lanthanide doped nanoscintillators featuring both XEA and XEOL can be used as a flexible detector to realize flat-panel-free X-ray imaging of 3D electronic objects (Nature, 2021, 590, 410). In our case, the enhancement in emission intensity for both the XEA and XEOL should improve the imaging quality. In addition, the core/shell/shell NPs can respond to X-rays, UV and NIR photons simultaneously, which provides a chance to develop broadband photodetectors.

Figure R1 Schematic illustration of applications of NPs featuring time-dependent XEA colour evolution for bio-sensing.

Considering that the length of the revised manuscript may be too long, this peer-review response will be provided online and only the following simplified contents have been added in the conclusion section of the revised manuscript.

These results not only provide a facile avenue for developing X-ray activated afterglow nanomaterials with superior performances, but also to build a general platform to modulate the emission colour in fluoride core/shell NPs, which may find promising applications in biomedicine (i.e., biosensing, bio-imaging and photodynamic therapy), advanced anti-counterfeiting, displays, optoelectronic devices (i.e., broadband photodectors) and potentially many others.

Comment: Incorporation of Na ions: the nominal ratio of [Na]/[RE] is significantly increased whereas the actual increase is very small (Table S1). How was the [Na] determined? Can such small increase in [Na] within the NPs indeed induce such strong XEA enhancement; any theoretical approach that would support this?

Response: The variation trends of [Na] concentration was first revealed by energy dispersive X-ray spectroscopy (EDS) and further determined by inductively coupled plasma-optical emission spectroscopy (ICP-OES). As shown in Figure R2, the EDS analysis reveals that the Na signal enhances gradually with an increase of the [Na]

concentration in the precursors solution. The weight ratio of the cations was measured by ICP-OES and the corresponding molar ratio was calculated (Table S1). The molar ratio is normalized so that $[Na] + [Lu] + [Gd] + [Tb] = 100\%$. Compared with the $[Na]/[RE]$ of 2.5, the $[Na]$ concentration increased to $\sim 8.67\%$, 14.6% and 19.7% with increasing $[Na]/[RE]$ to 5, 7.5 and 10, respectively.

Figure R2 EDS spectrum of the NaLuF₄: Gd/Tb NPs prepared with $[Na]/[RE]$ of 10.

The Rietveld XRD refinement, energy dispersive X-ray spectroscopy, inductively coupled plasma-optical emission spectroscopy and X-ray photoelectron spectroscopy results have been analyzed in our original manuscript, which indicates that interstitial Na⁺ ions are formed in the NaLuF₄ crystal structure. The interstitial Na⁺ ions not only decrease binding energies of the F⁻ ions, but also bring electrostatic interactions between the F⁻ and interstitial Na⁺ ions. Thus, in the case of high $[Na]/[RE]$ ratio, the anion Frenkel defects are easier to be formed upon X-ray irradiation via elastic collisions between large-momentum X-ray photons and small fluoride ions, benefiting the formation of traps. To further verify the proposed mechanism, the formation energy of anion Frenkel defects in the NaLuF₄ crystal structure was calculated based on the density functional theory (DFT) with the projector augmented plane-wave method. The crystal structures of the original 2 x 2 x 1 supercell of NaLuF₄ and optimized interstitial Na-doped NaLuF₄ are shown in Figure S22a-b. The anion Frenkel defect formation energies E_f for the dislocation of F⁻ ions into interstitial sites with different separation distances (0.5, 1.0, 1.5, 2.0

and 2.5 Å) were calculated. As shown in Figure S22c, after the incorporation of interstitial Na^+ ions, the E_f for the Frenkel defects with various separation distances are consistently reduced. Furthermore, the E_f values for different interstitial Na^+ sites were calculated by considering the relaxation of all atoms owing to the introducing of interstitial Na^+ ions and formation of Frenkel defects (Figure S22d-e). The calculated results indicated that the E_f values also reduce after the formation of interstitial Na^+ ions, irrespective of their locations.

Fig. S22 Schematic illustration of the Frenkel defects with different distances in the NaLuF_4 structure without (a) and with (b) intestinal Na^+ ions. c The corresponding calculated Frenkel defect formation energies. Formation of Frenkel defects in the NaLuF_4 structures without (d) and with (e) intestinal Na^+ ions at different sites under the consideration of relaxation of all atoms.

The amount of X-ray induced Frenkel defects (n_F) can be expressed by the following equation [ref: Bollmann, W., Gorlich, P., Hauk, W. & Mothes, H. Ionic

conduction of pure and doped CaF₂ and SrF₂ crystals. *Phys. Status Solidi A* **2**, 157-170 (1970)]:

$$n_F = \sqrt{N_I N_i} e^{-\frac{E_f}{2kT}}$$

where N_l and N_i are the number of F⁻ lattices and interstitial sites, respectively, k and T are Boltzmann constant and temperature, respectively. Thus a decrease of E_f can lead to an evident increase of n_F . In the above calculations, the interstitial Na⁺ ions concentration is set as 11.1%, which is similar to our ICP results. Thus, the incorporation of interstitial Na⁺ ions can indeed induce strong XEA enhancement in our case. Actually, in some previously reported literatures, (i.e., *Inorg. Chem.*, 2020, 59, 17906-17915; *Nanoscale*, 2012, 4, 779-784; *Nanoscale*, 2013, 5, 8084-8089; *J. Phys. Chem. C*, 2017, 121, 26, 14274-14284), doping inert ions with small concentration has shown to increase the upconversion or downshifting emission intensities as well. Similarly, it is reasonable to understand the enhancement of XEA intensity in our cases.

Fig. S22 and following contents have been added in the revised manuscript or supporting information.

Upon X-ray irradiation, both the reduced binding energy and introduced electrostatic interaction can promote the formation of anion Frenkel defects. Density functional theory (DFT) calculations revealed that the anion Frenkel defect formation energies (E_f) for the dislocation of F⁻ ions into interstitial sites with different separation distances (0.5, 1.0, 1.5, 2.0 and 2.5 Å) were consistently reduced after the formation of interstitial Na⁺ ions (Fig. S22a-c). Similar results were achieved upon changing the interstitial Na⁺ sites (Fig. S22d-e).

The amount of X-ray induced Frenkel defects (n_F) can be expressed by the following equation⁵:

$$n_F = \sqrt{N_I N_i} e^{-\frac{E_f}{2kT}}$$

where N_l and N_i are the number of F⁻ lattices and interstitial sites, respectively, k and T are Boltzmann constant and temperature, respectively. In this occasion, the decrease

of E_f can lead to an evident increase of n_F , which benefits the formation of high concentration traps.

Comment: What is the reason for the XEA loss at a ratio of 12.5?

Response: When fixing the $[F]/[RE]$ ratio at 3.75, the pure $NaLuF_4$ phase was achieved with the $[Na]/[RE]$ ratio changing from 2.5 to 10, while the impurity NaF phase emerged with the $[Na]/[RE]$ ratio further increasing to 12.5. In this case, the XEA intensity increased with increasing the $[Na]/[RE]$ ratio from 2.5 to 10 and then decreased with further increasing the $[Na]/[RE]$ ratio to 12.5. When fixing the $[F]/[RE]$ ratio at 5, the as-product remains the pure $NaLuF_4$ phase with increasing the $[Na]/[RE]$ ratio from 2.5 to 5, while the NaF phase emerges with further increasing the $[Na]/[RE]$ ratio from 7.5 to 12.5. For the $[F]/[RE]$ ratio being 3.75, a similar variation trend of XEA intensity against $[Na]/[RE]$ ratio has been confirmed. Moreover, when the impurity NaF phase was removed, the above XEA intensity variation trend was recorded as well (See Fig. S8 in the next comment). The results show that the XEA intensity was indeed increased after the removal of NaF (Figure R3 in the next comment). Based on these results, it is suggested that the existence of non-active NaF can reduce the total number of emitters so as to weaken the XEA intensity. Meanwhile, the high $[Na]/[RE]$ ratio induced excessive anion Frenkel defects can serve as quenching sites to decrease the XEA intensity via non-radiative relaxations.

The following contents have been added in the revised manuscript.

For the $[Na]/[RE]$ ratio above 10, the non-active NaF reduces the total number of emitters and the excessive anion Frenkel defects might promote non-radiative relaxation. Both effects contribute to the decrease of the XEOL and XEA intensities.

Comment: NaF formation: NaF can easily be washed away with a water-ethanol mixture. XEOL and XEA data were obtained on dry powders? If parts of these are indeed NaF , that may simply have an effect of how many X-ray-activated NPs there are and how many emitters compared to non-active NaF ? How do the samples perform after washing to remove NaF ?

Response: According to the reviewer’s helpful suggestions, the NaLuF₄: Gd/Tb NPs prepared with different [Na]/[RE] (2.5, 5, 7.5, 10 and 12.5) and [F]/[RE] of 5 were first treated with diluted HCl and then washed with water-ethanol mixture. In our original manuscript, we have clarified that the NaF phase emerged when the [Na]/[RE] ratio was 7.5, 10 or 12.5. As shown in Fig. S8, the excessive NaF (obtained with the conditions of [Na]/[RE] = 7.5, 10 and 12.5) was removed by washing with water-ethanol mixture. In this case, the XEA intensities of these pure phases without NaF were compared, where the XEOL/XEA data was obtained on dry powders. To estimate the ratio of emitters to non-active NaF and the corresponding influence on the XEA intensity, the EDS and XEA spectra of the NaLuF₄: Gd/Tb NPs ([Na]/[RE] of 10) before and after washing with water-ethanol were studied. Through comparing the [Na] content before and after washing with water-ethanol based on the EDS results, the NaF content was about 23 % for [Na]/[RE] = 10. The XEA intensity was indeed slightly increased after the removal of NaF (Figure R3). However, as shown in Fig. S8, the removal of NaF does not change the relationship between the XEA intensity and the [Na]/[RE] ratio. These results suggests that although the non-active NaF induced by [Na]/[RE] ratio high can reduce the total number of emitters so as to weaken the XEA intensity, high [Na]/[RE] ratio can contribute to the enhancement of XEA intensity in a more significant way by introducing the interstitial Na⁺.

Figure R3 XEA spectra of the NaLuF₄: Gd/Tb NPs ([Na]/[RE] of 10) before and after

the removal of NaF.

Fig. S8 XRD patterns (a) and XEA spectra (b) of the NaLuF₄: Gd/Tb NPs ([Na]/[RE] = (2.5, 5, 7.5, 10, and [F]/[RE] = 5) after washing with water-ethanol mixture.

The Fig. S8 and following content has been added in the revised manuscript.

The removal of NaF does not change the relationship between the XEA intensity and the [Na]/[RE] ratio, i.e., the XEA intensity increased with the [Na]/[RE] ratio increasing from 2.5 to 10 and then decreased when the [Na]/[RE] goes up to 12.5 (Fig. S8).

Comment: Can values for the afterglow (decay times) be given for comparison with phosphors reported in the literature (under comparable X-ray activation)? Are different afterglow time scales for various Ln due to the intrinsic Ln properties (references?) or due to the chosen mol%?

Response: According to the reviewer's helpful comment, the afterglow duration for the NaLuF₄:Tb and commercial SrAl₂O₄:Eu,Dy products were compared, and the XEA decay times for the NaLuF₄:Tb (5, 10, 15, 20 mol%), NaLuF₄:Dy (0.5, 1, 2 mol%), and NaLuF₄:Pr (0.5, 1, 2 mol%) NPs with different activators concentrations were systematically studied.

In our original manuscript, a spectrometer with PMT-CR131-TE detector (OmniFluo-Xray-JL) was used to test the XEA performances of those lanthanide doped fluoride NPs. To better measure the afterglow time of the present studied

NaLuF₄:Tb NPs, a better spectrometer (FLS980, Edinburgh Instrument) with R928 detector was employed. As shown in Fig. S14, under the same X-ray irradiation and measurement conditions, both the XEA intensity and afterglow time of the present studied NaLuF₄:Tb NPs are superior to that of commercial SrAl₂O₄:Eu,Dy persistent phosphor. The measured afterglow time of the NaLuF₄:Tb NPs was actually up to more than 6 hours.

Fig. S14 Compared XEA decay curves of the NaLuF₄: 15Gd/15Tb NPs and commercial SrAl₂O₄:Eu/Dy persistent phosphor (Edinburgh Instrument: FLS980 with R928 detector).

As shown in Fig. S27, the XEA decay profile is independent of Ln doping concentration for the Tb/Dy/Pr doped samples, indicating that the different afterglow time scales for various Ln³⁺ activators are mainly attributed to their intrinsic properties, instead of concentrations (mol%). Actually, in a recent literature (Light: Science & Applications, 2022, 11:51, Mechanism of the trivalent lanthanides' persistent luminescence in wide bandgap materials), the X-ray activated persistent luminescence from different Ln³⁺ activators were studied in detail. It is clarified that the intrinsic arrangement of the 4f electrons, anion coordination and cation substitution can influence the XEA performances of the trivalent lanthanide activators. In our case, since the parameters concerning either the anion coordination or cation

substitution are similar for different Lanthanide activators, their different XEA performances are mainly attributed to their different arrangement of the 4f electrons.

Fig. S27 XEOL spectra and XEA decay curves of different lanthanide activators, **a, d** Pr (0.5, 1, 2 mol%), **b, e** Dy (0.5, 1, 2 mol%), **c, f** Tb (5, 10, 15, 20 mol%).

The Fig. S14, Fig. S27 and the following contents have been added in the revised manuscript or supporting information.

The XEA intensity is stronger than that of previously reported $\text{Sr}_{13}\text{Al}_2\text{Si}_{10}\text{O}_{66}:\text{Eu}$ and commercial $\text{SrAl}_2\text{O}_4:\text{Eu}/\text{Dy}$, $\text{ZnS}:\text{Cu}$, $\text{ZnS}:\text{Cu}/\text{Co}$ and $\text{CaS}:\text{Eu}$ persistent phosphors (Fig. S13), and the afterglow time can last up to more than 6 hours (Fig. S14).

As shown in Fig. S27, the XEA decay profile is independent of the Ln doping concentration for the Tb/Dy/Pr doped samples, indicating that the different afterglow decay rates of Lanthanide activators are mainly ascribed to their different intrinsic arrangements of 4f electrons⁵², instead of concentrations.

Comment: Combination with various Ln (Figure 3): what is the role of Gd in the mechanism?

Response: In many previous reported Ce^{3+} sensitized downshifting and Yb^{3+} sensitized upconversion fluoride nanoparticles, such as $\text{NaYF}_4:\text{Ce},\text{Gd},\text{Tb},\text{Eu}$

(Nanoscale, 2013, 5, 9255), NaGdF₄:Eu (J. Mater. Chem. C, 2013, 1, 801), NaGdF₄:Tb/Eu (ACS Applied Materials & Interfaces 2017, 9, 31, 26184), NaGdF₄:Yb/Eu@NaGdF₄:Ce@NaGdF₄:Yb/Tb@NaYF₄ (Materials and Design, 2018, 152, 119), NaGdF₄:Yb/Tm@NaGdF₄:Eu (Adv. Mater., 22, 3266) and NaGdF₄:Yb/Tm@NaGdF₄:X (X= Tb, Eu, Dy, or Sm) (Nature Materials, 2011, 10, 968), it has been verified that the Gd³⁺ ions can promote the energy migration from Gd³⁺: ⁶P_{7/2} to Tb³⁺, Eu³⁺, Dy³⁺ and Sm³⁺ activators. Moreover, the ⁶P_{7/2} state and ground state of Gd³⁺ are separated by a relatively large energy gap (~3.2*10⁴ cm⁻¹), leading to a minimized energy loss caused by multiphonon emission and cross-relaxation (Science, 1999, 283, 663-666). As such, the Gd³⁺ ions have been widely employed to enhance luminescence emission intensities. In the X-ray excited optical luminescence field, it was verified that “at a low Gd³⁺ concentration, the excitation energy can be efficiently transferred from Gd³⁺ to Tb³⁺ activators. By comparison, at a high Gd³⁺ concentration, the excitation energy dissipates non-radiatively to quenching sites through energy migration, resulting in fast spontaneous emission of Tb³⁺ with low afterglow intensity. (Nature, 2021, 590, 410)”.

In our work, to further improve the XEA intensity via introducing energy migration process, the Gd³⁺ ions were codoped with various activators. As shown in Figure S9-10, both of the XEOL and XEA intensities were significantly increased with Gd³⁺ doping ratio of 15 mol%. Moreover, with increasing the Tb³⁺ ions concentration from 5 to 20 mol% in the NaLuF₄: 15Gd/Tb NPs (Figure S11), the Gd³⁺ emission at ~311 nm corresponding to its ⁶P_{7/2}→⁸S_{7/2} transition was evidently decreased gradually, which further verify the existence of energy transfer from Gd³⁺: ⁶P_{7/2} to Tb³⁺.

Based on the above discussion and additional experimental results, the following sentences were added in the “Spectroscopic study of afterglow intensification.” section of the revised manuscript.

With increasing the Tb³⁺ concentration, the Gd³⁺: ⁶P_{7/2}→⁸S_{7/2} emission intensity decreases (Fig. S11), revealing that Gd³⁺ can promote the energy migration from its ⁶P_{7/2} level to activators. Moreover, the relatively large energy gap (~3.2*10⁴ cm⁻¹)

between ${}^6P_{7/2}$ and the ground state of Gd^{3+} is in favor of minimizing energy loss caused by multiphonon assisted non-radiation relaxation and cross-relaxation⁴².

Fig. S11 XEOL spectra of the NaLuF₄: 15Gd/Tb NPs with different Tb³⁺ doping concentrations (5, 10, 15, and 20 mol%).

In addition, the references of (Nature Materials, 2011, 10, 968 and Adv. Mater., 22, 3266) were added along with the sentence of “It should be noted that the incorporation of Gd³⁺ ions with an optimal concentration of 15 mol% was used to facilitate the population of Tb³⁺ excited levels via energy transfer from Gd³⁺ to Tb³⁺ followed by the improved XEOL and XEA intensities (Fig. S8-9).”

Comment: Core-shell architectures: It is highlighted that XEA can be observed selectively from core or shell. Are core-shell structures truly needed? Would the same effect be observed when doping Ln ions for XEA together in the core? I am further wondering why the author chose to dope the UC emitters into the outer shell: this is the most sensitive location for upconverters given surface quenching; the core might have been the more intuitive region for UC dopants.

Response: Thanks for the reviewer’s kind comment. To reveal that the core-shell

structures are of significance for the realization of bright time-dependent multicolour evolution, the NaLuF₄:15Gd/15Tb/0.5Sm, NaLuF₄:15Gd/15Tb/0.5Sm@NaYF₄, NaLuF₄:15Gd/15Tb/0.5Pr, NaLuF₄:15Gd/15Tb/0.5Pr@NaYF₄, NaLuF₄:15Gd/0.5Sm/0.5Dy, NaLuF₄:15Gd/0.5Sm/0.5Dy@NaYF₄ NPs were prepared and studied.

As shown in Fig. S32a,d, when codoping 15Tb/0.5Sm in the NaLuF₄:15Gd core layer, both XEA intensities from the dual activators were not observed. Even after coating a NaYF₄ inert shell layer, the XEA intensity was still greatly weaker than that of 15Tb@0.5Sm. In this case, it is not clear to distinguish the time-dependent colour variations (Fig. S32d). As shown in Fig. S32b,d, compared with the NaLuF₄:15Tb@NaLuF₄:15Gd/0.5Pr, both XEA intensities of the NaLuF₄:15Gd/15Tb/0.5Pr and NaLuF₄:15Gd/15Tb/0.5Pr@NaYF₄ were evidently decreased, and there is no colour change over time. As shown in Fig. S32c-d, compared with the NaLuF₄:15Gd/0.5Sm@NaLuF₄:15Gd/0.5Dy, both XEA intensities of the NaLuF₄:15Gd/0.5Sm/0.5Dy and NaLuF₄:15Gd/0.5Sm/0.5Dy@NaYF₄ were greatly decreased. The fact that codoping dual activators in the core layer induced significant decrease in XEA intensity is mainly attributed to the strong, deleterious non-radiative cross-relaxations between dual activators. As a result, it is very important to construct a core@shell architecture to separate the different activators so as to realize strong XEA intensities as well as time-dependent colour evolution.

Fig. S32 **a** XEA spectra of the NaLuF₄:15Gd/15Tb/0.5Sm (Tb/Sm), NaLuF₄:15Gd/15Tb/0.5Sm@NaYF₄ (Tb/Sm@Y) and NaLuF₄:15Gd/15Tb@NaLuF₄:10Gd/0.5Sm (Tb@Sm) NPs. **b** XEA spectra of the NaLuF₄:15Gd/15Tb/0.5Pr (Tb/Pr), NaLuF₄:15Gd/15Tb/0.5Pr@NaYF₄ (Tb/Pr@Y) and NaLuF₄:15Tb@NaLuF₄:15Gd/0.5Pr (Tb@Pr). **c** XEA spectra of the NaLuF₄:15Gd/0.5Sm/0.5Dy (Sm/Dy), NaLuF₄:15Gd/0.5Sm/0.5Dy@NaYF₄ (Sm/Dy@Y) and NaLuF₄:15Tb@NaLuF₄:15Gd/0.5Dy (Sm@Dy) NPs. **d** XEA photographs of the above NPs with different delay times.

We agree that the upconverters are sensitive to surface quenchers, especially in the small NPs. In our original manuscript, the Tb@Ce/Sm@Yb/Tm core@shell@shell NPs were used as an example to show the colour editing process. Actually, the change of activators distributions in the core@shell@shell structure may lead to different colour variation processes. To reveal the influence of Yb/Tm location on the UC intensity, the Tb@Ce/Sm@Yb/Tm, Tb@Yb/Tm@Ce/Sm and Yb/Tm@Tb@Ce/Sm core@shell@shell NPs were prepared and studied. As shown in

Fig. S39, the UC intensity is enhanced when Yb/Tm are located in the core owing to surface passivation, while the XEA intensity is evidently decreased. The XEA intensity decreases over time, while the UC intensity is stable and can be tuned by changing the excitation power. In this case, a stronger initial intensity for XEA is in favor of the realization of multicolour evolution.

Fig. S39 UC (a) and XEA (b) spectra of the Yb/Tm@Tb@Ce/Sm, Tb@Yb/Tm@Ce/Sm and Tb@Ce/Sm@Yb/Tm NPs.

The Fig. S32, S39 and the following content have been added in the revised manuscript or supporting information.

The XEA intensities were greatly reduced when codoping dual-activators (Tb/Sm, Tb/Pr or Sm/Dy) in the core layer (Fig. S32), indicating the spatial separation of activators in a core@shell structure is of importance for the realization of bright multicolour evolution.

After exchanging the NaLuF₄:15Gd/15Tb and NaGdF₄:49Yb/1Tm layer in the core@shell@shell structure, the UC intensity was enhanced while the XEA intensity was greatly reduced (Fig. S39).

Comment: Size and size distributions: standard deviations should be provided. Shell thicknesses should be included.

Response: According to the reviewer's kind comment, the standard deviations of the size distributions of the core and shell thicknesses have been provided in Fig. S1 and

S24 in the revised supporting information.

Fig. S1 Histograms of size distributions of the NaLuF₄: Gd/Tb NPs prepared with different [Na]/[RE], 2.5 (a), 5 (b), 7.5 (c) and 10 (d).

Fig. S24 TEM images of the NaYF₄ core (a), NaYF₄@NaLuF₄: Gd/Tb NPs inert-core@active-shell NPs prepared with [Na]/[RE] of 2.5 (b) and 10 (c) in the shell layer. (d-f) are their corresponding histogram size distributions of the top face.

Comment: Figure panels: the figure panels are of high quality and attractive. I think

they could be further improved if adding labels for used Ln ions or concentrations to allow the reader to identify these critical parameters at a glance rather than looking for them in captions or text.

Response: According to the reviewer's kind suggestion, the used Ln ions and their concentrations have been added to the revised Fig. 2 and Fig. 4 in the revised manuscript.

Comment: What are the dimensions of the phosphor spots photographed (a scale bar would be helpful)? Are those photographs of dry powders or dispersions? At which time point where the photos taken?

Response: As suggested, the scale bar has been added in the revised Fig.2, Fig.4, Fig. 5 and Fig.6 in the revised manuscript. The circular photographs were recorded by loading the dry powders in a sample holder. After turning off the X-rays, the first photograph was recorded immediately (shorter than ~3 seconds) by iPhone.

Comment: Figure 2c: The linear lines may be misleading towards a linear correlation between ratio and intensity (which seems not to be the case based on intensities seen in Fig. 2b).

Response: According to the reviewer's kind suggestion, the linear lines shown in Figure 2c was replaced by histograms, which can clearly show the correlation between the ratio and the intensity for different Lanthanide activators. The revised Fig. 2 is shown below.

Fig. 2 Intensification of XEA via employing excessive Na⁺ precursors. XRD patterns (a) and XEA spectra (b) of the NaLuF₄: Gd/Tb NPs prepared with different [Na]/[RE] ratios. c Integral XEA intensities of the NaLuF₄: 15Gd/(0.5Pr, 15Tb, 0.5Dy or 0.5Sm) NPs prepared with [Na]/[RE] of 2.5 and 10. TEM images of the NaLuF₄: Gd/Tb NPs prepared with [Na]/[RE] of 2.5 (d), 5 (e), 7.5 (f) and 10 (g). Inset of e represents the three dimensional shape of a single NP. Inset of f shows the corresponding high-resolution TEM image. Photographs of those NPs doped with different activators at [Na]/[RE] of 2.5 (h) and 10 (i). X-ray operation was set at a voltage of 30 kV for 5 min.

Comment: Multi-colour modulation: How fast do these colour changes happen? Providing colour points at shorter time intervals may be helpful. Figure 4: Spectra as a function of time to show the contributions of the different Ln may be insightful.

Response: As suggested, the XEA at different delay time for the Tb@Sm, Tb@Pr or Sm@Dy NPs were studied. To reveal how fast the colour changes, more photographs with shorter time intervals have been recorded.

The Fig. S28, S30 and S31 have been added in the revised supporting information.

Fig. S28 XEA spectra and corresponding photographs with different delay times of the Tb@Sm NPs.

Fig. S30 XEA spectra and corresponding photographs with different delay times of the Tb@Pr NPs.

Fig. S31 XEA spectra and corresponding photographs with different delay times of the Sm@Dy NPs.

Comment: XEA / DS / UC combination: What was the rationale to suppress XEA from Sm? Was there any EXA seen from Tm (as in F. Zhang, Nature Nanotech 2021)?

Response: As shown in Fig. S38, after incorporating Ce³⁺ ions into the NaLuF₄ host, the XEA intensities of the Tb³⁺ and Sm³⁺ ions decrease. All the Ce³⁺, Gd³⁺ and Sm³⁺ ions can capture electrons from the traps, in another word, there exists a competition among them. Compared with the parity forbidden transitions within the f-manifold of lanthanides, the 4fⁿ-4fⁿ⁻¹5d¹ optical transitions are often characterized by a high radiative emission probability because the f-d transition is electrical-dipole allowed. In our case, the Ce³⁺ ion exhibits much larger absorption cross-section via 4f-5d transition than those of Gd³⁺ and Sm³⁺ ions via 4f-4f transitions. Thus, most of the electrons deposited in the traps will be captured by Ce³⁺ ions. Although the energy

transfer processes from Ce^{3+} to Gd^{3+} and then to Sm^{3+} has been widely used to produce DS emission, it is hard to avoid energy loss during the energy transfer from Ce^{3+} to Gd^{3+} and from Gd^{3+} to Sm^{3+} . For example, in previously reported Ce/Gd/Tb(Sm) codoped systems, the calculated best energy transfer efficiency from Ce to Tb (Sm) can be up to ~75% (Nanoscale, 2012, 4, 3450; Journal of the American Ceramic Society, 2017, 100, 2069-2080). However, the calculation formula of energy transfer efficiency η is based on the simplified equation $\eta = 1 - \tau_1/\tau_2$, where τ_1 and τ_2 are the decay times of donor in the presence and absence of the acceptor (Journal of Physics and Chemistry of Solids, 2003, 64, 841–846). The actual energy transfer efficiency should be lower than the calculated values owing to the existence of several other influence parameters. As a result, the incorporated Ce^{3+} ions will capture large number of electrons deposited in the traps, many of which are lost during the energy transfer processes. The scenario is different for Gd/Sm codoped ones. For the Gd^{3+} and Sm^{3+} ions via 4f-4f transitions, although the Gd^{3+} ions can capture part of electrons deposited in the traps, the improved electron population efficiency in Sm^{3+} excited levels (owing to the introduction of energy transfer from Gd^{3+} to Sm^{3+}) can remain, leading to the enhanced XEA. Thus, Ce^{3+} can suppress Sm^{3+} XEA.

To study the XEA of Tm in the NaLuF₄:15Gd/15Tb@NaLuF₄:15Gd/10Ce/0.5Sm@NaGdF₄:49Yb/1Tm, the NaGdF₄:1Tm, NaGdF₄:49Yb/1Tm, NaYF₄:1Tm and NaYF₄:49Yb/1Tm NPs were prepared. As shown in Fig. S38, the XEA intensities of the Tm³⁺ ions in both NaYF₄ and NaGdF₄ hosts decreased greatly after the incorporation of Yb³⁺ ions. The high concentration Yb³⁺ captures many electrons from the traps, and the efficiency of energy transfer from Yb³⁺ to Tm³⁺ is low in the XEA process (its mechanism is quite different from that of UC). Thus, the electron population in the Tm³⁺ is greatly reduced after the incorporation of Yb³⁺ ions. Therefore, in the NaLuF₄:15Gd/15Tb@NaLuF₄:15Gd/10Ce/0.5Sm@NaGdF₄:49Yb/1Tm NPs, the XEA intensity is very weak in the NaGdF₄:49Yb/1Tm layer as well.

Fig. S38 Compared XEA spectra of the NaGdF₄: 1Tm and NaGdF₄: 49Yb/1Tm (a), NaYF₄: 1Tm and NaYF₄: 49Yb/1Tm NPs (b).

The Fig. S37-S38 and following contents have been added in the revised manuscript or supporting information.

It should be noted that the Ce³⁺ ions were employed to absorb the 254 nm UV photons and inhibit Sm³⁺ XEA (Fig. S37), and the Yb³⁺ ions were used to absorb the 980 nm NIR photons and inhibit Tm³⁺ XEA (Fig. S38).

The Ce³⁺, Gd³⁺ and Sm³⁺ ions compete to capture electrons from the traps. Compared with the parity forbidden transitions within the f-manifold of lanthanides, the 4fⁿ-4fⁿ⁻¹5d¹ optical transitions are often characterized by high radiative emission probability because the f-d transition is electrical-dipole allowed. In our case, the Ce³⁺ ion exhibits much larger absorption cross-section via 4f-5d transition than those of Gd³⁺ and Sm³⁺ ions via 4f-4f transitions. Thus, most of the electrons deposited in the traps will be captured by Ce³⁺ ions. Although the energy transfer processes from Ce³⁺ to Gd³⁺ and then to Sm³⁺ has been widely used to produce DS emission, it is hard to avoid energy loss during the energy transfer from Ce³⁺ to Gd³⁺ and from Gd³⁺ to Sm³⁺. As a result, the incorporated Ce³⁺ ions will capture large number of electrons deposited in the traps, many of which are lost during the energy transfer processes.

The high concentration Yb³⁺ captures many electrons from the traps, and the efficiency of energy transfer from Yb³⁺ to Tm³⁺ is low for XEA process. Thus, the electron population in the Tm³⁺ is greatly reduced after the incorporation of

Yb³⁺ ions.

Comment: DS and UC spectra should be given to show which Ln ions are active as a function of time as well as E-level diagram showing the excitation and emission pathways.

Response: According to the reviewer's helpful suggestions, the DS and UC spectra as well as the E-level diagram were studied. The content "This bowknot presented medium orchid UC corresponding to the Tm³⁺ emission under 980 nm laser excitation, while it changed to sulfur yellow DS corresponding to the Tb³⁺ and Sm³⁺ emissions under UV illumination (Fig. 5b and Fig. S40)" and Fig. S40 have been added in the revised manuscript or supporting information.

Fig. S40 DS (a) and UC (b) emission spectra of the Tb@Ce/Sm@Yb/Tm core@shell@shell NPs. Proposed energy transfer processes of the DS (c) and UC (d).

Comment: Comparison of the performance of the nanophosphors with those reported in the literature is missing (while literature is discussed in the introduction, there is lack of discussion of the results in light of existing literature). I am further wondering how balanced the reference list is in terms of diversity of research groups and their geographical origin.

Response: According to the reviewer's helpful comment, we have compared the XEA performance of the NaLuF₄:Tb, Sr₁₃Al₂₂Si₁₀O₆₆:Eu, SrAl₂O₄:Eu/Dy, ZnS:Cu, ZnS:Cu/Co and ZnS:Eu. The commercial SrAl₂O₄:Eu/Dy, ZnS:Cu, ZnS:Cu/Co and ZnS:Eu persistent phosphors were purchased from Xiucan Chemical and Andron Technologies company. The Sr₁₃Al₂₂Si₁₀O₆₆:Eu is a persistent phosphor that was studied in our previous literature (*Journal of Advanced Ceramic*, 2022, 11, 974-983). In the introduction and the main text, we mentioned different kinds of long persistent phosphors including aluminates, silicates, sulfides and carbon dots. Because the afterglow time of most reported carbon dots is less than 10 seconds (Angew. Chem. Int. Edit., 2019, 131, 7356-7361; Nanoscale Adv., 2021, 3, 5053-5061; Angew. Chem. Int. Edit., 2021, 133, 22427-22433), we did not compare the XEA performance between carbon dots and NaLuF₄:Tb NPs. As shown in Fig. S13, the initial XEA intensity of the NaLuF₄:Tb NPs was much stronger than those conventional persistent phosphors upon the same X-ray irradiation conditions and PMT detector. Even after 20 mins, the XEA intensity of the NaLuF₄:Tb NPs remained much stronger than the others. These results are similar to the previous reported literature (*Nature*, 2021, 590, 410-415). Moreover, as mentioned in our manuscript that the fluoride NPs are much more appropriate for the integration of afterglow, DS and UC emissions via the construction of an appropriate core/shell nanoarchitecture. However, it is hard to simultaneously realize XEA, DS and UC emissions in those conventional persistent phosphors.

The Fig. S13 and the following contents have been added in the revised manuscript or supporting information.

The XEA intensity is stronger than previously reported Sr₁₃Al₂₂Si₁₀O₆₆:Eu and

commercial $\text{SrAl}_2\text{O}_4:\text{Eu}/\text{Dy}$, $\text{ZnS}:\text{Cu}$, $\text{ZnS}:\text{Cu}/\text{Co}$ and $\text{CaS}:\text{Eu}$ persistent phosphors (Fig. S13), and the afterglow time can last up to more than 6 hours (Fig. S14).

Fig. S13 a Normalized XEOL spectra of the $\text{NaLuF}_4:\text{Tb}$ NPs and various of conventional persistent phosphors. EDS spectrum (b) and SEM images (c) of different persistent phosphors. d XEA decay curves of those compared persistent phosphors. Inset shows their corresponding normalized initial XEA intensities.

The cited references for those conventional persistent phosphors are published by different research groups over the world including Germany, America, China, France, Singapore and Russia.

Comment: There are a few typos and sentences that may need clarification:

Response: Thanks for the reviewer's careful reading.

The "line 24 (the a)" was corrected to "the";

The "line 185 (produce)" was corrected to "benefits"

Comment: what is meant by "the afterglows exhibit same excitation and emission wavelengths"? Work by F. Zhang or B. Viana demonstrates examples of X-ray excited vis or NIR emission, clearly different excitation and emission wavelengths.

Response: We are sorry for misleading your understanding. Generally speaking, the emission intensity of persistent phosphor decreases with time when excitation stops, while the relative intensity between different emission peaks (if there are more than one emission peaks in the afterglow spectra) does not change with time. For the core/multi-shell fluoride nanostructure in the present study, however, the XEA decay profile for different activator varies and can be further tuned by adjusting the doping position. This can actually enable a controllable time-dependent variation in the relative XEA intensity between different activators, which is hardly achieved with traditional persistent phosphors. For better understanding, the mentioned sentence has been corrected to "For most previously reported long persistent phosphors, such as aluminates^{23,24}, silicates^{25,26}, sulfides^{27,28} and carbon dots^{29,30}, the afterglows exhibits a time-dependent intensity decrease but with a time-independent spectrum profile."

Comment: "time-dependent colour modulation on demand" – the colour change seems to be dependent on the decay of the X-ray induced afterglow. How can this be controlled on demand?

Response: To show that the time-dependent colour modulation can be controlled on demand, a schematic illustration is plotted to reveal the colour variations. As shown in Figure R4, the afterglow contains green and red colors (a), and the green one decays slower than the red one (b). In this case, if the initial intensity of the red colour is much stronger than the green one, then the output afterglow will be changed from red to green (c); if the initial intensity of the red colour is similar to the green one, then the output afterglow will be changed from yellow to green (d); if the initial intensity of the red colour is weaker than the green one, then the output afterglow will be green

(e). This is only a simplified illustration. In our case, the Tb (green), Dy (cyan) and Pr (red) activators exhibit different afterglow colours and different decay rates. Thus, through a combination of different activators and tuning their relative initial XEA intensities (i.e., tuning the [Na]/[RR] ratio and the Gd^{3+} doping content), the time-dependent colour change can be controlled on demand. Similarly, the DS and UC intensities can be tuned by the excitation power. Hence, for the core/shell/shell NPs that exhibit XEA, DS and UC emissions, the output colours can be modulated as well.

Figure R4 Schematic illustration of XEA colour variations at different conditions.

Comment: Figure S3: the caption should probably also include 12.5.

Response: The 12.5 has been added in the revised Fig. S3.

Comment: $Na_3HfF_7:Yb/Er$ and $NaLuF_4:Yb/Ho$ are mentioned – how were these obtained and what are their sizes / size distributions?

Response: According to the reviewer's kind comment, the preparation methods and their SEM images as well as size distributions for the $Na_3HfF_7:Yb/Er$ and $NaLuF_4:Yb/Ho$ products are provided in the revised manuscript. It should be noted

that the as-prepared Na_3HfF_7 : Yb/Er exhibited much broader size distributions (from ~120 to ~320 nm), which did not provide owing to the big error.

The following contents and Fig. S42 has been added in the revised supporting information.

Synthesis of lanthanide-doped Na_3HfF_7 : 8Yb/6Er NCs. Hafnium (IV) acetylacetonate (0.84 mmol), $\text{Yb}(\text{NO}_3)_3 \cdot 5\text{H}_2\text{O}$ (0.08 mmol), $\text{Er}(\text{NO}_3)_3 \cdot 5\text{H}_2\text{O}$ (0.06 mmol) and 20mL ethanol were added into a 50 mL beaker and stirred for 10 min. Then 12mL OA, 2.5mL OM and 2.5g NaOA were added into the above mixture with continuous stirring for another 20 min followed by the addition of 5mL deionized water containing 10 mmol NH_4F . After stirring at room temperature for about 30 min, the above solution was transferred into a 40 mL Teflon-lined autoclave, sealed and heated at 130 °C for 12 h. After the solution cooled down to room temperature, the product were precipitated via the addition of ethanol and collected via centrifugation.

Synthesis of NaLuF_4 :10Yb/10Ho core NPs. $\text{Lu}(\text{Ac})_3$ (0.64 mmol), $\text{Yb}(\text{Ac})_3$ (0.08 mmol) and $\text{Ho}(\text{Ac})_3$ (0.08 mmol) were added into a 50 mL three-necked bottle containing OA (8 mL). The mixture was heated at 150 °C for 30 min to remove water from the solution. Then 12 mL ODE was quickly added into the above solution and the resulted mixture was heated at 150 °C for another 30 min to form a transparent solution, and then cooled down to room temperature. Afterwards, 8 mL of methanol solution containing NH_4F (3 mmol) and NaOH (2 mmol) was added into the above solution and stirred at 60 °C for 40 min. After the methanol was fully evaporated, the solution was heated to 290 °C under N_2 atmosphere and kept for 90 min, and then cooled down to room temperature. The products were precipitated by centrifugation.

Fig. S42 SEM image (a) and EDS spectrum (b) of the $\text{Na}_3\text{HfF}_7:\text{Yb}/\text{Er}$. SEM image (a), EDS spectrum (b) and size distribution of the $\text{NaLuF}_4:\text{Yb}/\text{Ho}$.

Comment: I could not find information about instrumentation used for DS and UC.

Response: According to the reviewer's kind comment, the information about instrumentation used for DS and UC has been added in the revised manuscript.

The following sentence of “The DS and UC emission spectra were recorded with a spectrometer (Edinburgh FLS980) equipped with an adjustable laser diode (980 nm, 0 - 2 W) and Xenon lamp (450 W).” has been added in the Characterizations sections of the revised manuscript.

Reviewer 3:

Comment: This manuscript is concerned with the time dependent multicolour evolution of x-ray afterglow from lanthanide doped fluoride nanoparticles. This is mainly achieved by using different lanthanide ions as activators as they control the rate of afterglow decay. Then by having different compartments (core/shell) one has one part of the NP glowing in one colour, the other in another colour. By having the colour intensity decay at different time scales leads to a changing colour observed. This colour effects of the NPs can be further modified by illuminating them with NIR or UV light, in order to trigger upconversion (UC) or downshifting (DS), respectively. In general, this is interesting work and the authors also performed a comprehensive and careful characterization work on their nanoparticles and their photophysical properties. This was done for a systematic variation of the composition of the NPs, which is another strength of this article. Interesting here is certainly the observation that they can increase substantially the emission intensity by incorporating Na⁺ ions. In summary, a number of interesting optical properties are demonstrated in this work.

Response: We thank the reviewer's insightful comments and appreciate for the recognition of our results. We have revised the following comments appropriately.

Comment: One crucial point regarding the chosen architecture of these core-shell or core-shell-shell nanoparticles still must be explained. This concerns the explanation, why the construction of such rather complex NPs is necessary at all to arrive at the reported phenomena. In principle, many of them should also be achievable by simply mixing the individual NP with the materials contained in the core and the shell (or shells), isn't that the case. Therefore, the central point to be made is to explain what is the gain from having these different materials contained within one more complex NP that requires correspondingly more synthetic effort.

Response: According to the reviewer's kind comment, the XEA photograph of the mixture NPs is recorded. As shown in Fig. S33, when simply mixing the NaLuF₄:15Gd/15Tb and NaLuF₄:15Gd/0.5Sm NPs, the red, yellow and green colours can be observed simultaneously and separately, indicating the XEA colour is not

uniform.

Fig. S33 XEA photograph of the NaLuF₄:15Gd/15Tb and NaLuF₄:15Gd/0.5Sm mixture.

The Fig. S33 and the sentence of “It should be noted that the physical mixture of different NPs will lead to the non-uniform colour distribution (Fig. S33).” have been added in the revised manuscript or supporting information.

Moreover, the NaLuF₄:15Gd/15Tb/0.5Sm, NaLuF₄:15Gd/15Tb/0.5Sm@NaYF₄, NaLuF₄:15Gd/15Tb/0.5Pr, NaLuF₄:15Gd/15Tb/0.5Pr@NaYF₄, NaLuF₄:15Gd/0.5Sm/0.5Dy, NaLuF₄:15Gd/0.5Sm/0.5Dy@NaYF₄ NPs were studied to further reveal the importance of core-shell structures for the realization of time-dependent multicolour evolution.

As shown in Fig. S32a,d, when codoping 15Tb/0.5Sm in the NaLuF₄:15Gd core layer, both XEA intensities from the dual activators were not observed. Even after coating a NaYF₄ inert shell layer, the XEA intensity was still much weaker than that of 15Tb@0.5Sm. In this case, it is not easy to distinguish the time-dependent colour variations (Fig. S32d). As shown in Fig. S33b,d, compared with the NaLuF₄:15Tb@NaLuF₄:15Gd/0.5Pr, both XEA intensities of the NaLuF₄:15Gd/15Tb/0.5Pr and NaLuF₄:15Gd/15Tb/0.5Pr@NaYF₄ were evidently decreased, and there is no colour change over time. As shown in Fig. S32c-d, compared with the NaLuF₄:15Gd/0.5Sm@NaLuF₄:15Gd/0.5Dy, both XEA intensities of the NaLuF₄:15Gd/0.5Sm/0.5Dy and NaLuF₄:15Gd/0.5Sm/0.5Dy@NaYF₄ were greatly decreased. The fact that codoping dual activators in the core layer induced the greatly decrease of XEA intensity can be mainly attributed to strong deleterious non-radiative

cross-relaxations between dual activators. As a result, it is very important to construct a core@shell architecture to separate the different activators and then realize strong XEA intensities as well as time-dependent colour evolution.

Fig. S32 **a** XEA spectra of the NaLuF₄:15Gd/15Tb/0.5Sm (Tb/Sm), NaLuF₄:15Gd/15Tb/0.5Sm@NaYF₄ (Tb/Sm@Y) and NaLuF₄:15Gd/15Tb@NaLuF₄:10Gd/0.5Sm (Tb@Sm) NPs. **b** XEA spectra of the NaLuF₄:15Gd/15Tb/0.5Pr (Tb/Pr), NaLuF₄:15Gd/15Tb/0.5Pr@NaYF₄ (Tb/Pr@Y) and NaLuF₄:15Tb@NaLuF₄:15Gd/0.5Pr (Tb@Pr). **c** XEA spectra of the NaLuF₄:15Gd/0.5Sm/0.5Dy (Sm/Dy), NaLuF₄:15Gd/0.5Sm/0.5Dy@NaYF₄ (Sm/Dy@Y) and NaLuF₄:15Gd/0.5Sm@NaLuF₄:15Gd/0.5Dy (Sm@Dy) NPs. **d** XEA photographs of the above NPs with different delay times.

Comment: In Figs. 4 and 5 the change of the colours is shown visually. Interesting here could also be to see the full emission spectra as a function of time.

Response: Thanks for the reviewer's kind suggestions, the XEA at different delay

time for the Tb@Sm (Fig. S28), Tb@Pr (Fig. S30) and Sm@Dy (Fig. S31) NPs as well as UC (Fig. S41a) and DS (Fig. S41b) spectra at different times have been provided in the revised supporting information.

Fig. S28 XEA spectra and corresponding photographs with different delay times of the Tb@Sm NPs.

Fig. S30 XEA spectra and corresponding photographs with different delay times of the Tb@Pr NPs.

Fig. S31 XEA spectra and corresponding photographs with different delay times of the Sm@Dy NPs.

Fig. S41 Time-dependent UC (a) and DS (b) spectra of the pre-X-ray-irradiated bowknot gel.

Comment: Interesting would also to know how high is the percentage of the energy of the light emitted by the NPs compared to the energy put in via x-ray radiation.

Response: Thanks for the reviewer’s kind comment. Generally, the pulse height measurement method is accurate to determine the light yield of a scintillator, and the decay time should be several ns or shorter. However, the lifetimes of the trivalent lanthanide ions are generally up to several ms, which is hard to be measured by the pulse height measurement method. To show the scintillation intensity of the present studied NaLuF₄:Tb NPs, its XEOL intensity is compared with the commercial CsI:Tl and BGO scintillators. As shown in Fig. S12, the XEOL intensity of the NaLuF₄:Tb NPs is about 0.42 and 2.17 times the strength of the commercial CsI:Tl and BGO scintillators, respectively.

Fig. S12 Compared XEOL spectra and the corresponding normalized integral intensities of the NaLuF₄:Tb NPs, commercial CsI:Tl and BGO scintillators.

The Fig. S12 and sentence of “The XEOL intensity of the NaLuF₄:Tb NPs ([Na]/[RE] = 10) is about 0.42 and 2.17 times the strength of the commercial CsI:Tl and BGO scintillators, respectively (Fig. S12).” have been added in the revised manuscript.

Comment: As a minor point on page the units of the crystal lattice should be given.

Response: Thanks for pointing out. The unit (Å) of the crystal lattice has been added in the revised manuscript.

Comment: Finally, the use of the English language should be improved at various places, best by having this done by a native speaker.

Response: According to the reviewer's kind comment, we have checked the English language in the whole manuscript carefully and re-organized a few sentences appropriately in the revised manuscript.

Comment: In summary, I think that this manuscript has the potential for becoming published but the crucial point for that would be a convincing explanation, why the structure of the NPs employed here is superior to simply a mixture of individual NPs.

Response: We thank the reviewer's insightful comments again. The importance of the core/shell structure in our work has been studied and added in the revised manuscript or supporting information. We hope the reviewer agrees.

REVIEWER COMMENTS

Reviewer #1 (Remarks to the Author):

The revised version has been notably improved and now the paper is worth to be considered for publication.

Reviewer #2 (Remarks to the Author):

The revised version of the manuscript contains a significant amount of additional data and discussion, greatly contributing to the clarification of the questions that were previously brought up.

Still, I have some remaining questions with respect to nominal versus actual Na-RE ion ratio, and a few minor comments (see below).

“High” Na-RE ratios are found to provide better performance in terms of XEA. However, the actual ratio as determined by ICP and presented in Table S1 seems not to be very high. Comparing nominal and actual values, the excess ratio of 10 only yields an actual ratio of 1.3. Adding Na in 2.5x excess, a minimal increase to a ratio of 1.096 was determined.

1. How were the actual ratios in Table S1 determined? The table provides information about RE% (but only relative to each other, summing up to 100%), but no information about Na%. Both would be needed to follow the calculation of the actual ratio.
2. If I understand it correctly, the “pure” lattice would have a Na-RE ratio of 1. Thus, only up to max 0.3 interstitial Na ions per RE ion are found in the lattice. Why is so little of the excess Na incorporated into the lattice? How many interstitial Na ions would this be per unit cell?
3. What is the fate of the excess Na not incorporated? XRD patterns do not show the formation of NaF unless applying the highest Na-RE nominal ratios. Is any amorphous secondary phase formed?
4. Taking into account the actual Na-RE ion ratios, please explain why the specific values for distance in the DFT calculations were used (and which distances specifically are referred to).

Minor comments:

- Page 3, line 54: This sentence is difficult to understand. Do you mean something like maybe "...the afterglow exhibits a time-dependent intensity decrease, while the spectral profile remains unchanged."?
- Page 4, line 67: “An innovative route that incorporating interstitial...” – this sentence is difficult to understand.
- Figure 2c: What were the intensities normalized to? Were the integrated intensities obtained on the spectra shown in Figures S16-S18? Or did you use the images in h) and i) to extract the intensity/brightness (using some software).
- Page 8, line 148: each of the lattice parameters should have a unit (or Angstrom³).
- Page 12, line 207: The main text gives a particle size of ca. 43 nm, which seems not to be in agreement with the data show in the SI - average sizes given in Figure S24 are 60-65nm.
- Page 12, line 225: Lanthanide should be lanthanide
- I find the figure caption of Fig. 4 confusing with respect to core-only or core-shell architectures. Sample names that clearly provide info about the architecture (as already used elsewhere) would be helpful.
- A very large number of various samples was prepared and analyzed; yet, only a fraction of them is described in the synthesis. In case that various samples were synthesized using the same approach but different RE dopants in core or shell, this should be mentioned for clarity. A table summarizing the samples with their sizes and architectures as well as key XEA, DS and UC features might also be helpful (only in the SI).
- Potential application of the materials – on demand XEA: It is called "on demand". But once XEA is triggered, is there any way to control the color evolution? Can it be stopped / slowed down / accelerated? Or will it just happen over the sample-specific timeline? Does it depend on the environment of the NPs (solvent, temperature, ...)? – I am not asking for such experimental

assessment, but for how these probes are expected to work under real-life conditions; especially with respect to the highlighted “on demand”.

- Potential application of the materials – bioimaging and biosensing as well as drug release monitoring: Taking into account that the emitted light falls into the visible spectral region, bioimaging of thicker samples / in-vivo seems not to be feasible to me. The suggested PDT, in contrast, seems to be promising to me, given that the samples could be excited outside the body and the PDT take affect afterwards, once admitted to the body.
- Experimental Section: was the precursor mixture stirred to remove water under ambient pressure or vacuum?
- SI line 77: the amount of NaOA should be given in mol as for the other compounds.
- Size distribution histograms should include a y-axis to show how many NPs were considered in the size count.
- SI line 136: “NPs NPs”
- Figure S5a and b: These are data for the ratio of 10 or 12.5?
- Figures S9 and S10, b: what was the “normalized integral intensity” normalized to? One might expect a max value of 1 in case of a normalized data representation.
- Schematic illustrations of crystal lattices: addition of a color code for the different ions might be helpful.

Reviewer #3 (Remarks to the Author):

In my opinion the authors did a very good job in replying to the concerns raised by the reviewers. Accordingly, I would then suggest publication of this work.

Response to reviewer's comments

We greatly appreciate the reviewers' insightful comments which are very helpful for further improvement of our manuscript. In response to the valuable comments raised by the referees, we provide point-by-point responses along with the modifications (marked in blue) made in the revised manuscript.

Reviewer 2:

Comment: The revised version of the manuscript contains a significant amount of additional data and discussion, greatly contributing to the clarification of the questions that were previously brought up.

Still, I have some remaining questions with respect to nominal versus actual Na-RE ion ratio, and a few minor comments (see below).

“High” Na-RE ratios are found to provide better performance in terms of XEA. However, the actual ratio as determined by ICP and presented in Table S1 seems not to be very high. Comparing nominal and actual values, the excess ratio of 10 only yields an actual ratio of 1.3. Adding Na in 2.5x excess, a minimal increase to a ratio of 1.096 was determined.

Response: We thank the reviewer's insightful comments and appreciate for the recognition of our results. We have revised the following comments appropriately.

Comment: How were the actual ratios in Table S1 determined? The table provides information about RE% (but only relative to each other, summing up to 100%), but no information about Na%. Both would be needed to follow the calculation of the actual ratio.

Response: The weight ratios of the cations were measured by ICP-OES, and then the corresponding molar ratio results were calculated (Table S1). To clearly reveal the Gd and Tb doping concentrations through replacing Lu in the NaLuF₄ host, the molar ratio of [RE] is normalized ($[Lu] + [Gd] + [Tb] = 100\%$). Then the [Na] ratio was calculated by $[Na]/([Lu] + [Gd] + [Tb])$.

The revised Table S1 has been added in the revised supporting information.

Table S1 Nominal and ICP-OES results of the NaLuF₄: Gd/Tb NPs prepared with different [Na]/[RE]. [RE] = [Lu] + [Gd] + [Tb] = 100%.

Nominal	ICP-OES results (mol %)				Actual
[Na]/[RE]	[Lu]	[Gd]	[Tb]	[Na]	[Na]/[RE]
2.5	69.9%	16.4%	13.7%	109.6%	1.096
5	68.8%	17.6%	13.6%	119.1%	1.191
7.5	70.5%	15.1%	14.4%	125.6%	1.256
10	68.0%	17.4%	14.6%	131.2%	1.312

Comment: If I understand it correctly, the “pure” lattice would have a Na-RE ratio of 1. Thus, only up to max 0.3 interstitial Na ions per RE ion are found in the lattice. Why is so little of the excess Na incorporated into the lattice? How many interstitial Na ions would this be per unit cell?

Response: We agree with that the “pure” lattice has a Na-RE ratio of 1, and up to max 0.3 interstitial Na ions per RE ion is found in the lattice in our case. Firstly, most Na is preserved in the final reaction solution (See the next comment), indicating that most of them only participate in the growth of NPs, but not incorporated into the final NPs; secondly, the incorporation of intestinal Na ions probably requires a high [Na] concentration in the reaction solution and needs to overcome a high energy barrier. As a result, only a small amount of the excess Na was incorporated into the lattice. There are three Lu atoms per unit cell in the employed NaLuF₄ crystal structure. Thus, for the case of [Na]/[RE] = 10, there is about 0.9 interstitial Na ion in an unit cell.

Comment: What is the fate of the excess Na not incorporated? XRD patterns do not show the formation of NaF unless applying the highest Na-RE nominal ratios. Is any amorphous secondary phase formed?

Response: To reveal the fate of the excess Na not incorporated, the composition of the discarded solution after precipitation and centrifugation for the NaLuF₄: Gd/Tb NPs ([Na]/[RE]) = 10 was measured by EDS and elements mapping. As shown in

Fig. R1a-b, the EDS spectrum revealed that there existed a large amount of Na element but only trace lanthanides, indicating that most of [Na] precursors only participate in the growth of NPs, but not incorporated into the final NPs. The strong Na signal in the EDS mapping result (Fig. R1c-d) further reveals the existence of most Na ions in the discarded solution.

Fig. R1 EDS spectrum (a), elements ratio (b), SEM and mapping result of the discarded solution after centrifugation for the NaLuF₄: Gd/Tb NPs ([Na]/[RE]) = 10.

As revealed in the Fig. S5 and S6, for the case of [F]/[RE] = 3.75, the secondary NaF phase is formed, with the [Na]/[RE] ratio up to 12.5; while for the case of [F]/[RE] = 5, the secondary NaF phase is formed when the [Na]/[RE] ratio is up to 7.5. In another word, because most Na is preserved in the reaction solution, the secondary NaF phase is not formed when the [Na]/[RE] ratio is lower than 10 and [F]/[RE] = 3.75.

Comment: Taking into account the actual Na-RE ion ratios, please explain why the specific values for distance in the DFT calculations were used (and which distances specifically are referred to).

Response: In a previously reported literature (Ou, X. et al. High-resolution X-ray

luminescence extension imaging. Nature, 2021, 590, 410-415), the first-principles calculations based on density functional theory was used to monitor the structural relaxation of anion Frenkel pairs at various distances. They found that “interstitial fluoride ions gradually diffuse back to original vacancies when the proximity of these two subdefects is less than 3 Å. For defect pairs with a larger separation (more than 3 Å), interstitial fluoride ions can be stabilized due to increased energy barriers, except under stimulation with heating or light exposure.”. Considering the XEA properties in our case are repeatable, the anion Frenkel defect formation energies E_f for the dislocation of F^- ions into interstitial sites with different distances (0.5, 1.0, 1.5, 2.0 and 2.5 Å) were calculated.

Comment: Page 3, line 54: This sentence is difficult to understand. Do you mean something like maybe "...the afterglow exhibits a time-dependent intensity decrease, while the spectral profile remains unchanged."?

Response: This explanation is exactly what we want to describe. The original sentence has been corrected to “For most previously reported long persistent phosphors, such as aluminates^{23,24}, silicates^{25,26}, sulfides^{27,28} and carbon dots^{29,30}, the afterglows exhibit a time-dependent intensity decrease, while the spectral profiles remain unchanged.”

Comment: Page 4, line 67: “An innovative route that incorporating interstitial...” – this sentence is difficult to understand.

Response: The original sentence of “An innovative route that incorporating interstitial Na^+ ions inside the nanocrystal structure was employed to amplify the XEA intensities of the $NaLuF_4$: Gd/(Pr, Tb, Dy or Sm) NPs.” is corrected to “Incorporating interstitial Na^+ ions inside the nanocrystal structure was employed as an innovative route to amplify the XEA intensities of the $NaLuF_4$: Gd/(Pr, Tb, Dy or Sm) NPs.” in the revised manuscript.

Comment: Figure 2c: What were the intensities normalized to? Were the integrated

intensities obtained on the spectra shown in Figures S16-S18? Or did you use the images in h) and i) to extract the intensity/brightness (using some software).

Response: The integrated intensities were obtained in the spectra shown in Figures S3 and S16-S18. To clearly show the degree of XEOL intensification of the Pr, Tb, Dy and Sm activators after the incorporation of interstitial Na ions, the XEOL intensities for all the $[Na]/[RE] = 2.5$ were normalized to 1. It should be noted that Figure 2c cannot be used to compare relative XEOL intensities among different lanthanide activators.

The sentence of “The XEOL intensities for the $[Na]/[RE] = 2.5$ were normalized to 1.” has been added in Fig. 2c.

Comment: Page 8, line 148: each of the lattice parameters should have a unit (or Angstrom³).

Response: The original sentence has been corrected to “Rietveld XRD refinement results revealed that the crystal lattice increased from $5.9694 \text{ \AA} * 5.9694 \text{ \AA} * 3.5026 \text{ \AA}$ to $5.9762 \text{ \AA} * 5.9762 \text{ \AA} * 3.5154 \text{ \AA}$ (Fig. S19).” in the revised manuscript.

Comment: Page 12, line 207: The main text gives a particle size of ca. 43 nm, which seems not to be in agreement with the data show in the SI - average sizes given in Figure S24 are 60-65nm.

Response: The mean particle size shown in the main text corresponding to the Figure S24 has been corrected to $\sim 62 \pm 2$ nm.

Comment: Page 12, line 225: Lanthanide should be lanthanide.

Response: Lanthanide shown in Page 12, line 225 has been corrected to lanthanide.

Comment: I find the figure caption of Fig. 4 confusing with respect to core-only or core-shell architectures. Sample names that clearly provide info about the architecture (as already used elsewhere) would be helpful.

Response: As suggested, the “core-only” has been added after the NaLuF₄:

15Gd/(0.5Pr, 15Tb, 0.5Dy or 0.5Sm) NPs in the caption of Fig. 4.

In addition, for the paragraph of “As an example of practical multidimensional display application,...”, the core-only NPs and core@shell@shell NPs were both emerged in the text, thus, the “**core-only**” has been added accordingly as well.

Comment: A very large number of various samples was prepared and analyzed; yet, only a fraction of them is described in the synthesis. In case that various samples were synthesized using the same approach but different RE dopants in core or shell, this should be mentioned for clarity. A table summarizing the samples with their sizes and architectures as well as key XEA, DS and UC features might also be helpful (only in the SI).

Response: Thanks for the reviewer’s kind comment. The following contents have been added in the revised supporting information.

The *NaLuF₄:15Gd/0.5Pr*, *NaLuF₄:15Gd/0.5Dy* and *NaLuF₄:15Gd/0.5Sm* NPs were prepared via similar experimental procedures, except with different doping activators.

The *NaLuF₄:15Gd/15Tb@NaYF₄*, *NaYF₄@NaLuF₄:15Gd/15Tb*, *NaLuF₄:15Gd/15Tb@NaLuF₄:15Gd/0.5Sm*, *NaLuF₄:15Gd/15Tb@NaLuF₄:15Gd/0.5Pr*, *NaLuF₄:15Gd/0.5Sm@NaLuF₄:15Gd/0.5Dy* were prepared via similar experimental procedures except using different doping activators and [Na]/[RE] ratio.

The *NaLuF₄:49Yb/1Tm* NPs were prepared via similar experimental procedures, except using different doping activators.

The comparison of different samples studied in this work is shown below.

Sample	[Na]/[RE]	Size (nm)	Architecture	Emission mode	Emission color
NaLuF₄:15Gd/0.5Pr	10	/	Core-only	XEA	Pink
NaLuF₄:15Gd/15Tb	10	47	Core-only	XEA	Green
NaLuF₄:15Gd/0.5Dy	10	/	Core-only	XEA	Cyan
NaLuF₄:15Gd/0.5Sm	10	/	Core-only	XEA	Red
NaLuF₄:15Gd/15Tb@	Core 10	/	Core@shell	XEA	Green

NaYF ₄	Shell 2.5				
NaYF ₄	2.5	49.5	Core-only	/	/
NaYF ₄ @NaLuF ₄ :15Gd/15Tb	Core 2.5 Shell 2.5	61.5	Core@shell	XEA	Green
NaYF ₄ @NaLuF ₄ :15Gd/15Tb	Core 2.5 Shell 10	64.5	Core@shell	XEA	Green
NaLuF ₄ :15Gd/15Tb@ NaLuF ₄ :15Gd/0.5Sm	Core 10 Shell 10	/	Core@shell	XEA	Pale yellow → Green
NaLuF ₄ :15Gd/15Tb@ NaLuF ₄ :15Gd/0.5Pr	Core 10 Shell 10	/	Core@shell	XEA	Yellow → Green
NaLuF ₄ :15Gd/0.5Sm@ NaLuF ₄ :15Gd/0.5Dy	Core 10 Shell 10	60	Core@shell	XEA	Turquoise → Dark turquoise → Green
NaLuF ₄ :15Gd/15Tb @NaLuF ₄ :15Gd/10C e/0.5Sm@NaGdF ₄ :49 Yb/1Tm	Core 10 Shell1 2.5 Shell2 2.5	90	Core@shell @shell	XEA	Green
				DC	Orange
				UC	Purple
NaLuF ₄ :10Yb/10Ho	2.5	31	Core-only	UC	Yellow
NaLuF ₄ :49Yb/1Tm	2.5	/	Core-only	UC	Purple

Comment: Potential application of the materials – on demand XEA: It is called "on demand". But once XEA is triggered, is there any way to control the color evolution? Can it be stopped / slowed down / accelerated? Or will it just happen over the sample-specific timeline? Does it depend on the environment of the NPs (solvent, temperature, ...)? – I am not asking for such experimental assessment, but for how these probes are expected to work under real-life conditions; especially with respect to the highlighted “on demand”.

Response: As shown in our previous response letter, we would like to emphasize that the time-dependent colour modulation can be controlled on demand before excitation. We can design the time-dependent colour evolution process by tuning the chemical compositions or by using a core@shell structure for specific requirements of different applications. The UC and DS are generally generated under NIR and UV pumping sources, respectively. Thus, when combining the XEA with UC and DS, after triggering the XEA, the time-dependent colour evolution process can be controlled by changing the excitation wavelength or power (NIR and UV).

Considering the afterglow decay rate can be accelerated by elevating temperature, the time-dependent colour evolution processes will be influenced by temperature. Similarly, the other environmental parameters, which can disturb the behavior of electrons releasing from traps, might influence the time-dependent colour evolution processes as well. As a result, it is of significance to further study these issues for practical applications.

The previously related respond contents are copy below.

To show that the time-dependent colour modulation can be controlled on demand, a schematic illustration is plotted revealing the colour variations. As shown in Figure R2, the afterglow contains green and red colors (a), and the green one decays slower than the red one (b). In this case, if the initial intensity of the red colour is much stronger than the green one, then the output afterglow will be changed from red to green (c); if the initial intensity of the red colour is similar to the green one, then the output afterglow will be changed from yellow to green (d); if the initial intensity of the red colour is weaker than the green one, then the output afterglow will be green (e). This is only a simplified illustration. In our case, the Tb (green), Dy (cyan) and Pr (red) activators exhibit different afterglow colours and different decay rates. Thus, through a combination of different activators and tuning their relative initial XEA intensities (i.e., tuning the [Na]/[RR] ratio and the Gd³⁺ doping content), the time-dependent colour change can be controlled on demand. Similarly, the DS and UC intensities can be tuned by the excitation power. Hence, for the core/shell/shell NPs that exhibit XEA, DS and UC emissions, the output colours can be modulated as well.

Figure R2 Schematic illustration of XEA colour variations at different conditions.

Comment: Potential application of the materials – bioimaging and biosensing as well as drug release monitoring: Taking into account that the emitted light falls into the visible spectral region, bioimaging of thicker samples / in-vivo seems not to be feasible to me. The suggested PDT, in contrast, seems to be promising to me, given that the samples could be excited outside the body and the PDT take affect afterwards, once admitted to the body.

Response: According to the reviewer’s helpful suggestion, the examples of bioimaging and biosensing are removed from the main text. The original sentence has been corrected to “...which may find promising applications in biomedicine (i.e., photodynamic thereapy)”.

Comment: Experimental Section: was the precursor mixture stirred to remove water under ambient pressure or vacuum?

Response: Pure N_2 was used throughout for the preparation of NPs. At 150 °C, water was removed together with the flowing N_2 .

The following content has been added in the Experimental Section.

The mixture was heated at 150 °C for 30 min to remove water from the solution under N₂ atmosphere.

Comment: SI line 77: the amount of NaOA should be given in mol as for the other compounds.

Response: The “NaOA (8.21 mmol)” has been added in the Experimental Section.

Comment: Size distribution histograms should include a y-axis to show how many NPs were considered in the size count.

Response: As suggested, the Counts have been added on the y-axis in the size distribution histograms in Figures S1, S24 and S42.

Fig. S1 Histograms of size distributions of the NaLuF₄: Gd/Tb NPs prepared with different [Na]/[RE], 2.5 (a), 5 (b), 7.5 (c) and 10 (d).

Fig. S24 TEM images of the NaYF₄ core-only (a), NaYF₄@NaLuF₄: Gd/Tb NPs inert-core@active-shell NPs prepared with [Na]/[RE] of 2.5 (b) and 10 (c) in the shell layer. (d-f) are their corresponding histogram size distributions of the top face.

Fig. S42 SEM image (a) and EDS spectrum (b) of the Na₃HfF₇:Yb/Er. SEM image (c), EDS spectrum (d) and size distribution of the NaLuF₄:Yb/Ho.

Comment: SI line 136: “NPs NPs”

Response: Thanks for the reviewer’s careful reading. The repeated NPs has been removed in the revised supporting information.

Comment: Figure S5a and b: These are data for the ratio of 10 or 12.5?

Response: Figure S5a and b are are data for the ratio of 12.5. The captions shown in Fig. S5 has been corrected to “**Fig. S5** XRD pattern (a), TEM image (b) of the NaLuF₄: Gd/Tb NPs prepared with [Na]/[RE] of 12.5. XEOL (c) and XEA (d) of the NaLuF₄: Gd/Tb NPs prepared with [Na]/[RE] of 10 and 12.5. The JCPDS 361455 and 270726 represent the standard data of cubic NaF and hexagonal NaLuF₄.” in the revised supporting information.

Comment: Figures S9 and S10, b: what was the “normalized integral intensity” normalized to? One might expect a max value of 1 in case of a normalized data representation.

Response: The integral intensity of the NaLuF₄: Tb NPs (without Gd) was normalized to 1 in the Figures S9 and S10.

The sentence of “**The case of 0 mol% is normalized to 1.**” has been added in the captions of Figures S9 and S10.

Comment: Schematic illustrations of crystal lattices: addition of a color code for the different ions might be helpful.

Response: As suggested, the colour code of Na has been changed to another colour code in the Figure 3d.

REVIEWERS' COMMENTS

Reviewer #2 (Remarks to the Author):

The careful revision by the authors has further clarified the manuscript.

ICP data in Table S1 confirm nicely the relative amounts of Lu, Gd and Tb relative to each other, confirming a close match with the anticipated dopant % for Gd and Tb of 15%.

Here, I am wondering why the Na/RE ratio is calculated assuming a 100% RE concentration for each sample. One would expect slight changes in the ppm (or mol/L) of Gd, Tb and Lu from sample to sample. Why not dividing mol/L of Na by mol/L of RE (data as obtained by ICP)? Maybe such calculations were considered to obtain the shown [Na] relative to 100% RE. If so, okay.

Response to reviewer's comments

We greatly appreciate the reviewers' insightful comments which are very helpful for further improvement of our manuscript. In response to the valuable comments raised by the referees, we provide point-by-point responses along with the modifications made in the revised manuscript.

Reviewer 2:

Comment: The careful revision by the authors has further clarified the manuscript.

ICP data in Table S1 confirm nicely the relative amounts of Lu, Gd and Tb relative to each other, confirming a close match with the anticipated dopant % for Gd and Tb of 15%.

Here, I am wondering why the Na/RE ratio is calculated assuming a 100% RE concentration for each sample. One would expect slight changes in the ppm (or mol/L) of Gd, Tb and Lu from sample to sample. Why not dividing mol/L of Na by mol/L of RE (data as obtained by ICP)? Maybe such calculations were considered to obtain the shown [Na] relative to 100% RE. If so, okay.

Response: We thank the reviewer's insightful comments and appreciate for the recognition of our results.

Although the relative contents of Gd, Tb and Lu may change from sample to sample, this work highlights the influence of [Na]/[RE] ratio on the XEA intensities of lanthanide activators doped fluoride NPs. Thus, the calculation results are used to show the relative [Na] content to 100% RE.